# The Geometry of Algorithmic Stability: A Hodge Theoretic View on Structural vs. Statistical Instability

**Karen Sargsyan**  *karen.sarkisyan@gmail.com*
*Institute of Chemistry, Academia Sinica*

**Reviewed on OpenReview:** *https://openreview.net/forum?id=rFqsgVXZYO*

## Abstract

Algorithmic stability—the robustness of predictions to training data perturbations—is fundamental to reliable machine learning. We propose a unified mathematical framework that rigorously distinguishes between two sources of instability: structural inconsistency and statistical variance. We formalize structural inconsistency using Combinatorial Hodge Theory, characterizing it as cyclical flows (Condorcet cycles) on a graph of hypotheses. This framework reveals that methods like inflated operators and regularization specifically target these structural obstructions, while methods like bagging primarily address statistical variance. We provide direct empirical validation through three key experiments. First, in a controlled setting with engineered Condorcet cycles (pure structural instability), inflated operators achieve perfect stability while bagging fails, confirming the core distinction. Second, we validate on a standard digit classification task that structural obstructions are negligible ($||C_{cycle}|| \approx 2.3 \times 10^{-16}$, machine precision), explaining the empirical dominance of variance-reduction methods. Third, we demonstrate that significant structural obstructions naturally emerge in fairness-constrained model selection on real-world data ($||C_{cycle}|| = 0.857$, approximately $10^{15}$ times larger), providing a topological characterization of the instability arising from incompatible objectives.

## 1 Introduction

The generalization performance of machine learning models is deeply connected to their stability—the robustness of outputs to perturbations in the training data (Bousquet & Elisseeff, 2002). Stability matters not only for training algorithms but also for any data-driven selection procedure: choosing among pre-trained models, selecting from ensembles, or aggregating predictions. Methods to enhance stability have been developed across these contexts, including bagging (Breiman, 1996; Soloff et al., 2024), regularization (Bousquet & Elisseeff, 2002), and more recently, "inflated" operators (Adrian et al., 2024; Liang et al., 2025).

While effective, these methods appear disconnected, and it is often unclear which method is appropriate for a given source of instability. Moreover, the classical stability framework assumes i.i.d. data from a single distribution, but modern applications face distribution shift, domain adaptation, and multi-objective learning where this assumption fails. This fragmentation raises a fundamental question: Can we rigorously distinguish the sources of instability and unify the approaches to address them across both single- and multi-distributional settings?

We argue the answer is yes, provided by the tools of Combinatorial Hodge Theory applied to the geometry of preferences. We propose that instability arises from two distinct sources: statistical variance (sensitivity of the learning process to noise or near-ties) and structural inconsistency (fundamental conflicts in the data preferences). We can conceptualize structural inconsistency by viewing subsets of the data as "voters" and the potential models as "candidates."

**Example 1.1** (Running Example: The Ranking Cycle)**.** Consider selecting the best model among three candidates {A,B,C}. Suppose one subset of the data (voter group 1) strongly suggests A is better than B

($A > B$). A second subset suggests $B > C$, and a third suggests $C > A$. This creates a cycle: $A > B > C > A$. No single global ranking can satisfy all local preferences. An algorithm forced to choose a single winner will be highly unstable in this scenario.

This scenario is the well-known Condorcet Paradox. We demonstrate that these paradoxes, which can be rigorously characterized by the cyclical component in a Hodge decomposition (Lim, 2020; Jiang et al., 2011), represent structural instability. This leverages established connections between social choice theory and algebraic topology (Chichilnisky, 1980; Baryshnikov, 1993).

While we find that structural inconsistencies are rare in standard supervised learning tasks—explaining the empirical success of methods like bagging—we demonstrate that they arise naturally in domains with conflicting objectives, such as fairness-aware machine learning. Our framework provides a mathematical characterization of the inherent instability in these scenarios.

Our framework formalizes the stability problem as one of preference aggregation under metric constraints. We show that:

1. The Hodge decomposition of ordinal preferences characterizes the mathematical conditions that determine the source of instability. A non-zero cyclical component ($C_{cycle} > 0$) indicates structural inconsistency, while $C_{cycle} \approx 0$ indicates instability is primarily due to statistical variance.

2. Structural inconsistency requires structural solutions. Inflated operators provide obstruction resolution (target space enlargement), while regularization can provide obstruction prevention.

3. Statistical variance is effectively addressed by variance reduction techniques like bagging. Crucially, we demonstrate that bagging is ineffective against pure structural instability.

4. We demonstrate that the framework naturally generalizes beyond the standard single-distribution setting to encompass distribution shift, domain adaptation, and multi-objective learning. The standard algorithmic stability framework (i.i.d. samples from a single distribution) emerges as a special case—the restriction of the preference sheaf to a single point in distribution space. This generalization provides a rigorous mathematical foundation for understanding instability arising from heterogeneous data sources or conflicting distributional objectives.

This framework provides mathematical unification, revealing deep connections between algorithmic stability, social choice theory, and algebraic topology, and provides theoretical insight into selecting appropriate stabilization methods.

## 2 Related Work

Our work sits at the intersection of algorithmic stability, topological data analysis, and social choice theory.

### 2.1 Foundations of Algorithmic Stability

The foundational work of Bousquet & Elisseeff (2002) established the link between algorithmic stability and generalization. The standard paradigm often addresses stability through statistical variance reduction, canonically exemplified by bagging (Breiman, 1996), whose assumption-free stability properties have been recently formalized (Soloff et al., 2024).

More recently, research has focused on the instability inherent in selection procedures like the argmax. Inflated operators (Adrian et al., 2024; Liang et al., 2025) address this by returning a stable set of near-optimal solutions, motivated largely by the metric sensitivity of the loss function under near-ties. Our work provides a unifying theoretical framework that reinterprets these methods as strategies for handling topological obstructions in the underlying data preferences.

## 2.2 Topological Data Analysis in Machine Learning

The application of Topological Data Analysis (TDA) to machine learning is a growing field ((Wasserman, 2018; Hensel et al., 2021; Zia et al., 2024)). Much of this work focuses on the geometry of the parameter space or the loss landscape. For instance, persistent homology and Morse theory have been used to identify obstructions to optimization, such as the structure of local minima and saddle points (Barannikov et al., 2020).

Crucially, our work analyzes the geometry of the preference space—how data induces rankings over the hypothesis space. The obstructions we identify are not features of the optimization landscape, but rather fundamental inconsistencies in the problem definition itself (conflicting data preferences), which exist independently of the optimization procedure.

## 2.3 Cohomology and Hodge Theory in ML

The mathematical tools we employ have precursors in related ML domains, though their application and interpretation differ significantly from ours.

**Fairness.** The foundational theory of topological social choice (Chichilnisky, 1980; Baryshnikov, 1993) has been applied to diagnose impossibility results in fairness. Prior work demonstrates that simultaneously satisfying disparate fairness criteria can be impossible due to Condorcet-like cycles across different metrics. Our work generalizes this insight from fairness to the foundational property of algorithmic stability.

**Rank Aggregation.** Combinatorial Hodge theory was introduced to the problem of rank aggregation by Lim (2020); Jiang et al. (2011). They utilized the Hodge decomposition to extract a global ranking (the gradient component) from pairwise comparison data and quantify the remaining inconsistencies (the cyclical components). While this provides a powerful tool for analyzing ranking data, our work makes the crucial connection between this mathematical structure and algorithmic stability. We reinterpret the cyclical component as the obstruction that causes instability, thereby providing a unifying framework for understanding stability-enhancing methods like bagging and inflated operators.

# 3 Stability as Constrained Preference Aggregation

We formalize the learning problem and stability constraints, analyzing how local data preferences can conflict.

## 3.1 The Learning Setup and Stability

Let $\mathcal{X}$ be the data space and let $\mathcal{P}$ denote a family of probability distributions over $\mathcal{X}$. We define the dataset space $\mathcal{D}$ as the space of all training datasets of fixed size $N$ that may be encountered in practice, potentially arising from different distributions in $\mathcal{P}$.

**Distributional Structure.** In the most general setting, $\mathcal{D}$ encompasses datasets sampled from a mixture or family of distributions, representing scenarios such as distribution shift, domain adaptation, or heterogeneous data sources. We equip $\mathcal{D}$ with an adjacency structure: two datasets $S, S' \in \mathcal{D}$ are *adjacent* if they are sufficiently similar in the distributional sense. This can be formalized via:

- **Point-wise adjacency:** $S$ and $S'$ differ by exactly one data point (standard leave-one-out).

- **Distributional adjacency:** The empirical measures $\hat{P}_S$ and $\hat{P}_{S'}$ satisfy $W_p(\hat{P}_S, \hat{P}_{S'}) < \delta$ for some Wasserstein distance $W_p$ and tolerance $\delta > 0$.

**The Standard Case as Specialization.** When $\mathcal{P} = \{P_0\}$ consists of a single distribution and datasets are sampled i.i.d. from $P_0$, point-wise adjacency coincides with the classical leave-one-out perturbation. The algorithmic stability literature (Bousquet & Elisseeff, 2002) primarily focuses on this special case. Our framework naturally generalizes to accommodate the multi-distributional setting, treating the standard case as the restriction of the sheaf to a single point (see Section 4.5).

Throughout, we write $\mathcal{D}$ to denote the support of the dataset distribution, equipped with a prescribed probability measure $\mu$ over this distributional mixture. Expectations are taken with respect to $\mu$.

Let $\mathcal{H}$ be the hypothesis space, equipped with a metric $d_\mathcal{H}$. Let $L : \mathcal{H} \times \mathcal{D} \to R$ be the loss function. A learning algorithm $\mathcal{A} : \mathcal{D} \to \mathcal{H}$ aims to select $\mathcal{A}(S) \approx \arg\min_{h \in \mathcal{H}} L(h, S)$.

**Definition 3.1** (Algorithmic Stability)**.** An algorithm $\mathcal{A}$ is $\epsilon$-stable if for any two adjacent datasets $S, S' \in \mathcal{D}$: $d_\mathcal{H}(\mathcal{A}(S), \mathcal{A}(S')) \leq \epsilon$.

### 3.2 Local Preferences and the Accuracy-Stability Tradeoff

The loss function induces preferences over the hypothesis space for each dataset.

**Definition 3.2** (Induced Preference Relation)**.** For a dataset $S$, we say $h_1$ is preferred over $h_2$ (denoted $h_1 >_S h_2$) if $L(h_1, S) < L(h_2, S)$.

To capture the set of viable hypotheses, we define the set of near-optimal solutions.

**Definition 3.3** (Near-Optimal Hypothesis Set)**.** For accuracy tolerance $\delta \geq 0$, the $\delta$-optimal hypothesis set for dataset $S$ is: $O_\delta(S) := \{h \in \mathcal{H} | L(h, S) \leq \min_{h' \in \mathcal{H}} L(h', S) + \delta\}$.

The fundamental challenge is finding an algorithm that is simultaneously accurate and stable.

**Definition 3.4** (The Stable Selection Problem)**.** Given $\epsilon, \delta \geq 0$, the Stable Selection Problem is to find a map $\mathcal{A} : \mathcal{D} \to \mathcal{H}$ such that for all $S, S' \in \mathcal{D}$:

1. $\mathcal{A}(S) \in O_\delta(S)$ (Accuracy)

2. If $S, S'$ are adjacent, $d_\mathcal{H}(\mathcal{A}(S), \mathcal{A}(S')) \leq \epsilon$ (Stability)

To ensure the problem is well-posed, we assume standard regularity conditions.

**Assumption 3.5.** The dataset space $\mathcal{D}$ (as defined by adjacency) is connected.

This assumption is necessary to analyze the global implications of local stability constraints. If $\mathcal{D}$ were disconnected, an algorithm could be locally stable within each component but globally unstable across components. For the definition of $\mathcal{D}$ given above (fixed size $N$, differing by one point), this connectivity generally holds.

## 4 Cyclical Obstructions to Stability

We introduce the mathematical framework used to characterize inconsistencies in preferences. While motivated by deep connections between social choice theory and abstract algebraic topology (Chichilnisky, 1980; Baryshnikov, 1993, see Appendix A), we utilize the concrete machinery of Combinatorial Hodge Theory (Lim, 2020; Jiang et al., 2011). This approach analyzes flows on a graph, providing a rigorous and accessible way to quantify structural inconsistencies (cycles) in the data preferences.

### 4.1 The Graph of Hypotheses and Preference Flows

We focus on a finite subset of relevant hypotheses $\mathcal{H}_k = \{h_1, \dots, h_k\}$ (e.g., models in an ensemble or candidates in model selection). We construct a complete graph $K$ where the vertices are the hypotheses $\mathcal{H}_k$.

We analyze the preferences induced by the data over these hypotheses. As introduced earlier, we view the collection of datasets $\mathcal{D}$ as "voters." In practical contexts, such as analyzing an existing ensemble on a test set (as in Experiment 2), individual data points can also serve as the voters, providing an empirical estimate of the preferences.

**Definition 4.1** (Preference Profile)**.** A preference profile $P$ is a collection of preference relations $\{>_S\}_{S \in \mathcal{D}}$ over $\mathcal{H}_k$.

We are interested in how these preferences manifest as flows on the edges of the graph $K$.

**Definition 4.2** (Cochains/Flows)**.**

- A 0-cochain is a function assigning values to vertices (e.g., a loss or utility function).

- A 1-cochain (or flow) is a function assigning values to edges (e.g., pairwise comparisons).

The fundamental operator connecting these is the coboundary operator $d^0$. It transforms a 0-cochain (potential) $f$ into a 1-cochain (flow): $(d^0 f)(h_i, h_j) = f(h_j) - f(h_i)$.

**Definition 4.3** (Gradient Flow)**.** A flow $C$ is a *gradient flow* (or coboundary) if it can be derived from a global potential function, i.e., $C = d^0 f$. Gradient flows represent perfectly consistent preferences corresponding to a global ranking.

### 4.2 Aggregating Preferences: The Importance of Ordinal Data

To analyze stability, we must aggregate the local preferences in the profile $P$ into a single global flow $C_P$ on the graph $K$. The method of aggregation is critical.

**Cardinal Aggregation.** If we aggregate the losses (cardinal values) directly, e.g., by averaging the loss differences across datasets: $C_P^{card}(h_i, h_j) = E_{S \sim \mathcal{D}}[L(h_j, S) - L(h_i, S)]$. This construction *always* results in a gradient flow, as it is the gradient of the aggregate loss function $L_{agg}(h) = E_S[L(h, S)]$. Therefore, $C_P^{card}$ cannot detect Condorcet cycles. To see this explicitly, define $L_{agg}(h) := E_S[L(h, S)]$. Then:

$$\begin{aligned} C_P^{card}(h_i, h_j) &= E_S[L(h_j, S) - L(h_i, S)] \\ &= L_{agg}(h_j) - L_{agg}(h_i) = (d^0 L_{agg})(h_i, h_j). \end{aligned}$$

Thus $C_P^{card} = d^0 L_{agg}$ is a gradient flow by construction.

**Ordinal Aggregation (Pairwise Majority Vote).** To detect structural inconsistencies analogous to those in social choice theory, we must aggregate the rankings (ordinal preferences). We define the Ordinal Preference Flow $C_P^{ord}$ using Pairwise Majority Vote (PMV): $C_P^{ord}(h_i, h_j) = P_{S \sim \mathcal{D}}(h_i >_S h_j) - P_{S \sim \mathcal{D}}(h_j >_S h_i)$. This represents the net preference for $h_i$ over $h_j$ across the dataset distribution. Crucially, $C_P^{ord}$ is generally *not* a gradient flow.

**Aggregation Across Distributions.** In the general distributional setting where datasets in $\mathcal{D}$ may arise from different source distributions $P \in \mathcal{P}$, the ordinal aggregation via PMV naturally captures conflicts between *distributional preferences.*

Consider a scenario where different subregions of the data space are governed by different distributions (e.g., different demographic groups in fairness applications, different domains in transfer learning, or different wilderness areas in ecological classification). Each distribution $P_i$ induces local preferences over hypotheses. The aggregated flow $C_P^{ord}$ synthesizes these potentially conflicting local preferences into a global structure.

When datasets from distribution $P_i$ consistently prefer $h_a$ over $h_b$, while datasets from distribution $P_j$ consistently prefer $h_b$ over $h_a$, the ordinal aggregation reveals this structural conflict. If such conflicts form cycles across multiple distributions, the resulting $C_P^{ord}$ exhibits a non-zero cyclical component, indicating that no single hypothesis can simultaneously satisfy the preferences induced by all distributions in $\mathcal{P}$.

Formally, we can decompose the preference flow by distribution:

$$C_P^{ord}(h_i, h_j) = \sum_{P \in \mathcal{P}} w_P \cdot C_P^{ord}(h_i, h_j) \tag{1}$$

where $w_P$ is the weight (prevalence) of distribution $P$ in the mixture, and $C_P^{ord}$ is the preference flow induced by datasets sampled from $P$ alone. Cyclical obstructions arise when the local flows $\{C_P^{ord}\}_P$ are mutually inconsistent in their global aggregation.

**Empirical estimation:** In practice, we estimate the preference profile using a test set. Individual test points $x$ reveal the preference structure induced by training on different data samples: $C_P^{ord}(h_i, h_j) \approx$

$\frac{1}{n} \sum_{x \in X}[1_{\ell(h_i,x)<\ell(h_j,x)} - 1_{\ell(h_j,x)<\ell(h_i,x)}]$, where $\ell(h,x)$ is the pointwise loss and 1 denotes the indicator function. This empirically estimates the aggregate preference relationship between hypotheses.

## 4.3 Hodge Decomposition and Cyclical Obstructions

We use the Hodge decomposition theorem to analyze the structure of the ordinal preference flow $C_P^{ord}$. The divergence operator $\delta^1 : C^1 \rightarrow C^0$ is the adjoint of $d^0$ with respect to the standard inner product. A flow $C$ is *divergence-free* (or harmonic) if $\delta^1 C = 0$, meaning it has zero net flow at each vertex.

**Theorem 4.4** (Hodge Decomposition on Graphs). *Any flow $C$ on the graph $K$ can be uniquely and orthogonally decomposed as: $C = C_{gradient} + C_{cycle}$. Where $C_{gradient}$ is a gradient flow (the consistent part), and $C_{cycle}$ is a divergence-free flow (the cyclical part).*

(See Appendix A.4 for mathematical details and the definition of the divergence operator).

This theorem provides the central tool for our framework. It allows us to quantify the degree to which a preference profile deviates from a global consensus.

**Definition 4.5** (Cyclical Obstruction). The component $C_{cycle}$ in the Hodge decomposition of the ordinal preference flow $C_P^{ord}$ is the *cyclical obstruction*. It mathematically characterizes the structural inconsistency (Condorcet cycles) inherent in the preference profile $P$.

The magnitude $||C_{cycle}||$ quantifies the severity of the obstruction. When $||C_{cycle}|| = 0$, the preferences are perfectly consistent. When $||C_{cycle}|| > 0$, a structural inconsistency exists. In the language of algebraic topology, $C_{cycle}$ corresponds to a non-trivial element in the first cohomology group ($H^1$). We use the standard $L^2$ norm: for a 1-cochain $C$, $||C|| = \sqrt{\sum_{(i,j)\in \text{edges}} C(h_i, h_j)^2}$.

**Example 4.6** (The Condorcet Cocycle). The profile $(A > B, B > C, C > A)$ from Example 1.1, if dominant across the data, results in a $C_P^{ord}$ where the flow is purely cyclical ($C_{gradient} = 0$). This is the canonical example of a structural obstruction.

## 4.4 Mapping Obstructions to Instability

We now connect this mathematical structure back to the Stable Selection Problem. While the existence of a cyclical obstruction ($||C_{cycle}|| > 0$) indicates inconsistency, instability arises when this inconsistency is sufficiently strong and localized.

**Definition 4.7** (Strong Structural Inconsistency). A preference profile $P$ exhibits a strong structural inconsistency with margin $\gamma > 0$ if there exists a cycle of hypotheses $\{h_1, \ldots, h_k\}$ and disjoint regions of the dataset space, $\mathcal{D}_i \subseteq \mathcal{D}$, such that for $S \in \mathcal{D}_i$, $h_i$ is strongly preferred over $h_{i+1}$ (indices modulo k): $L(h_i, S) < L(h_{i+1}, S) - \gamma$.

**Remark 4.8.** Strong structural inconsistency (Definition 4.7) is a sufficient condition for non-zero cyclical obstruction when the cycle is sufficiently prevalent in the dataset distribution. Specifically, when disjoint regions $\mathcal{D}_i$ with the cycle structure induce net positive flow $C_P^{ord}(h_i, h_{i+1}) > 0$ for each edge $(h_i, h_{i+1})$ in the cycle, the ordinal aggregation via PMV necessarily yields $||C_{\text{cycle}}|| > 0$.

*Justification:* If $C_P^{ord}(h_i, h_{i+1}) > 0$ for all edges in the cycle, then summing around the cycle gives $\sum_{i=1}^{k} C_P^{ord}(h_i, h_{i+1}) > 0$. However, any gradient flow $d^0 f$ satisfies

$$\sum_{i=1}^{k} (d^0 f)(h_i, h_{i+1}) = \sum_{i=1}^{k} [f(h_{i+1}) - f(h_i)] = 0 \tag{2}$$

because each value $f(h_j)$ appears exactly once with a positive sign (when $j = i+1$) and once with a negative sign (when $j = i$), causing all terms to cancel. This contradiction implies $C_P^{ord}$ is not a gradient flow, forcing $||C_{\text{cycle}}|| > 0$ by the Hodge decomposition.

This condition implies that different regions of the data space have strongly conflicting preferences, forcing an accurate algorithm to choose different hypotheses in different regions.

**Proposition 4.9** (Instability Induced by Strong Obstructions). *If the preference profile induced by the data contains a strong structural inconsistency with margin $\gamma$, and for sufficiently small $\delta$ the accuracy constraint implies metric localization (i.e., there exists $\delta'(\delta) \to 0$ such that $d_{\mathcal{H}}(\mathcal{A}(S), h_i) < \delta'$ when $S \in \mathcal{D}_i$), then for $\delta < \gamma/2$ and sufficiently small $\epsilon$, the Stable Selection Problem is unsolvable.*

*Proof.* The condition of strong structural inconsistency ensures a separation margin $\gamma$. We choose the accuracy tolerance $\delta < \gamma/2$. If $S \in \mathcal{D}_i$, assuming $h_i$ is near-optimal, $L(h_i, S) < L(h_{i+1}, S) - \gamma$. Since $\gamma > 2\delta$, $h_{i+1}$ cannot be in the near-optimal set: $h_{i+1} \notin O_{\delta}(S)$. This confirms that the optimal sets in different regions are disjoint with respect to the hypotheses in the cycle.

To satisfy the accuracy constraint, an algorithm $\mathcal{A}$ must select $\mathcal{A}(S) \in O_{\delta}(S)$. The structural inconsistency forces the algorithm to choose between distinct hypotheses in different regions. We demonstrate that stability prevents this.

**Intuitive Case: Discrete Metric.** Consider the simplified case where $\mathcal{H}$ has the discrete metric ($d(h_a, h_b) = 1$ if $a \neq b$) and $\epsilon < 1$. Stability implies $\mathcal{A}(S) = \mathcal{A}(S')$ for adjacent $S, S'$. Since $\mathcal{D}$ is connected (Assumption 3.5), this forces a single global choice $h^*$. However, the strong structural inconsistency (a cycle) implies that for any $h^*$, there exists a region $\mathcal{D}_i$ where $h^*$ is strongly dispreferred and thus $h^* \notin O_{\delta}(S)$ for $S \in \mathcal{D}_i$. Thus, the problem is unsolvable.

**General Case: General Metric Space.** We generalize the argument using the triangle inequality. This requires that the structural inconsistency manifests metrically (standard regularity conditions assumed in Section 3.2). [1]

Let $h_i$ and $h_j$ be two distinct hypotheses in the cycle, separated by distance $D = d_{\mathcal{H}}(h_i, h_j) > 0$. Consider the corresponding regions $\mathcal{D}_i$ and $\mathcal{D}_j$. We assume that for sufficiently small $\delta$, the accuracy constraint implies metric localization. Formally, we assume there exists a $\delta'$ (which decreases as $\delta$ decreases) such that for $S \in \mathcal{D}_i$, $d_{\mathcal{H}}(\mathcal{A}(S), h_i) < \delta'$, and for $S' \in \mathcal{D}_j$, $d_{\mathcal{H}}(\mathcal{A}(S'), h_j) < \delta'$. We choose $\delta$ small enough such that $2\delta' < D/2$.

Since $\mathcal{D}$ is connected, there exists a finite path of adjacent datasets $S_0, S_1, \ldots, S_M$, where $S_0 \in \mathcal{D}_i$ and $S_M \in \mathcal{D}_j$.

By the stability constraint, $d_{\mathcal{H}}(\mathcal{A}(S_k), \mathcal{A}(S_{k+1})) \leq \epsilon$. By the triangle inequality along the path:

$$d_{\mathcal{H}}(\mathcal{A}(S_0), \mathcal{A}(S_M)) \leq \sum_{k=0}^{M-1} d_{\mathcal{H}}(\mathcal{A}(S_k), \mathcal{A}(S_{k+1})) \leq M \cdot \epsilon.$$

Now we relate this to the separation $D$ using the triangle inequality again:

$$D = d_{\mathcal{H}}(h_i, h_j) \leq d_{\mathcal{H}}(h_i, \mathcal{A}(S_0)) + d_{\mathcal{H}}(\mathcal{A}(S_0), \mathcal{A}(S_M)) + d_{\mathcal{H}}(\mathcal{A}(S_M), h_j)$$
$$< \delta' + (M \cdot \epsilon) + \delta'.$$

This gives the requirement $D < 2\delta' + M \cdot \epsilon$.

The distance $D$ and path length $M$ are fixed by the problem structure. We have chosen $\delta$ such that $2\delta' < D/2$. We can now choose $\epsilon$ sufficiently small such that $M \cdot \epsilon \leq D/2$. Then $2\delta' + M \cdot \epsilon < D$. This violates the derived inequality, creating a contradiction. Thus, the Stable Selection Problem is unsolvable. $\square$

When a strong cyclical obstruction exists, no single-valued algorithm can simultaneously satisfy the inconsistent preferences and the metric constraints of stability.

---

[1]Metric localization follows from standard continuity assumptions, such as $\beta$-Lipschitz continuity of the loss function in the hypothesis space: $|L(h, S) - L(h', S)| \leq \beta \cdot d_{\mathcal{H}}(h, h')$.

### 4.5 Sheaf-Theoretic Interpretation: Local-to-Global Obstructions

The distributional framework admits an elegant interpretation through the lens of sheaf theory, revealing the preference structure as a sheaf over distribution space and cyclical obstructions as cohomological obstructions to gluing local sections.

**The Preference Sheaf.** Let $(\mathcal{P}, \tau)$ be the space of distributions over $\mathcal{X}$, equipped with a suitable topology $\tau$ (e.g., the weak topology or topology induced by the Wasserstein metric). We define the *preference sheaf* $\mathscr{F}$ over $\mathcal{P}$ as follows:

- For each open set $U \subseteq \mathcal{P}$, the space of sections $\mathscr{F}(U)$ consists of hypothesis selection rules $\sigma : \bigcup_{P \in U} \mathcal{D}_P \to \mathcal{H}$ that are locally optimal and stable within $U$.

- For each point $P \in \mathcal{P}$, the stalk $\mathscr{F}_P$ represents the space of stable selections for datasets drawn from distribution $P$ alone, characterized by the local preference structure $C_P^{ord}$.

- Restriction maps $\rho_{UV} : \mathscr{F}(U) \to \mathscr{F}(V)$ for $V \subseteq U$ restrict the selection rule to the smaller region.

**Local Consistency, Global Obstruction.** A *global section* $\sigma \in \mathscr{F}(\mathcal{P})$ would be a hypothesis selection rule that is simultaneously accurate and stable across *all* distributions in $\mathcal{P}$. The existence of such a global section is precisely the solvability of the Stable Selection Problem (Definition 3.4) in the distributional setting.

The cyclical obstruction $C_{cycle} \neq 0$ is the manifestation of a cohomological obstruction: the local sections (stable selections within each distribution $P$) cannot be glued into a global section due to incompatible transition functions on overlaps. In classical sheaf cohomology, such obstructions are measured by cohomology groups $H^1(\mathcal{P}, \mathscr{F})$.

**The Standard Case.** When $\mathcal{P} = \{P_0\}$ is a single point, the sheaf theory trivializes: there is only one stalk, and the cohomological obstruction reduces to the local structure within that single distribution. This recovers the standard algorithmic stability setting. The power of the sheaf perspective emerges when $\mathcal{P}$ is non-trivial, capturing scenarios with genuine distribution shift or heterogeneity.

**Remark 4.10.** This sheaf-theoretic interpretation provides a rigorous mathematical foundation for understanding why certain learning problems admit stable solutions (global sections exist) while others do not (cohomological obstructions). It also suggests natural generalizations: higher-order cohomology groups $H^q(\mathcal{P}, \mathscr{F})$ for $q \geq 2$ may characterize more complex multi-way conflicts beyond pairwise cycles, a direction for future investigation.

## 5 A Mathematical Explanation for Stability Methods

Our framework, centered on the Hodge decomposition, rigorously distinguishes between two sources of instability and unifies the methods used to address them based on the presence or absence of structural obstructions ($C_{cycle}$).

### 5.1 Addressing Structural Obstructions ($C_{cycle} \neq 0$)

When the data preferences contain fundamental inconsistencies, the instability is structural. This requires methods that either resolve the ambiguity or enforce consistency.

#### 5.1.1 Obstruction Resolution: Target Space Enlargement

When obstructions are inherent ($C_{cycle} \neq 0$), a stable single-valued solution may not exist (Proposition 4.9). The mathematical solution is to change the target space.

In social choice theory, when a single consensus winner does not exist due to cycles, the solution is to identify a consensus *set*, such as the Top Trading Cycle (TTC) set—the minimal set of hypotheses $H^*$ that dominate those outside $H^*$. Inflated operators, rigorously analyzed by Adrian et al. (2024) and Liang et al. (2025),

implement this principle by enlarging the target space from $\mathcal{H}$ to $\mathcal{P}(\mathcal{H})$. For set-valued outputs, we use the Hausdorff distance on $\mathcal{P}(\mathcal{H})$:

$$d_H(A, B) = \max \left\{ \sup_{a \in A} \inf_{b \in B} d_{\mathcal{H}}(a, b), \sup_{b \in B} \inf_{a \in A} d_{\mathcal{H}}(a, b) \right\}$$

Alternatively, stability can be defined via set intersection: two set-valued outputs are stable if $O_\delta(S) \cap O_\delta(S') \neq \emptyset$.

**Definition 5.1** (Inflated Argmax (Adrian et al., 2024)). $argmax_\epsilon(w) := \{j | w_j \geq max_k w_k - \epsilon\}$.

**Interpretation 5.2. Cohomological Significance of Inflation** *Inflated operators resolve cyclical obstructions by systematic target space enlargement. The set-valued output of the inflated operator approximates the Top Trading Cycle set associated with the underlying preference cycle.*

**Argument.** Consider a profile $P$ with a significant cyclical obstruction, represented by a cycle $h_1 > \cdots > h_k > h_1$. As shown in Proposition 4.9, if this obstruction is strong, it leads to the failure of the single-valued Stable Selection Problem.

The Top Trading Cycle (TTC) set associated with this cycle is $H^* = \{h_1, ..., h_k\}$. Now consider the application of the inflated operator (e.g., returning the near-optimal set $O_\delta(S)$). If for each pair $h_i, h_j \in H^*$, the losses satisfy $|L(h_i, S) - L(h_j, S)| \leq \epsilon'$ uniformly across datasets $S$ in the relevant region (cycle balanced), then the losses of the hypotheses in the cycle will be close across the relevant region of the dataset space: $|L(h_i, S) - L(h_j, S)| \leq \epsilon'$ for $h_i, h_j \in H^*$.

If we choose the tolerance parameter $\delta \geq \epsilon'$, the near-optimal set $O_\delta(S)$ will contain $H^*$. If hypotheses outside the cycle have significantly higher loss, then $O_\delta(S) \approx H^*$.

The inflated operator thus identifies the set of hypotheses involved in the structural ambiguity characterized by the obstruction. Under bounded loss variations (a standard regularity assumption), for adjacent datasets $S, S'$ and a balanced cycle, if $h_i \in H^*$ has $L(h_i, S) \leq \min_h L(h, S) + \delta$, then by continuity $L(h_i, S') \leq \min_h L(h, S') + \delta'$ for $\delta' \approx \delta$. Thus all hypotheses in the cycle remain near-optimal on both datasets, guaranteeing $H^* \subseteq O_\delta(S) \cap O_\delta(S')$.

### 5.1.2 Obstruction Prevention: Enforcing Consistency

Alternatively, we can attempt to prevent obstructions by enforcing consistency during the learning process. Regularization-based ensemble stability adds penalty terms that discourage disagreement (e.g., $\mathcal{L}_{glue} = \sum_x \sum_{i,j} ||h_i(x) - h_j(x)||^2$, where $h_i(x)$ denotes the prediction of model $h_i$ on input $x$, and the sum is taken over a some discrete set of inputs on which agreement is required and all pairs of predictors). This explicitly enforces consensus among the ensemble members. By minimizing $\mathcal{L}_{glue}$, the optimization procedure forces the local preferences to align. When models are forced to produce similar outputs, their rankings across the data become more consistent, directly minimizing the inconsistencies that contribute to the cyclical component $C_{cycle}$. Our proposed method in Section 6.1 is a higher-order generalization of this approach.

### 5.2 Addressing Statistical Variance ($C_{cycle} \approx 0$)

When the cyclical component is near zero, the aggregated preferences are globally consistent. However, instability can still occur due to statistical variance—the sensitivity of the learning algorithm or the selection procedure (like argmax) to noise, near-ties, or specific data samples.

### 5.2.1 Bagging: Variance Reduction

Bagging (Breiman, 1996) utilizes bootstrap samples $\{S_i\}$ from $S$ and aggregates the results. Its effectiveness is statistically grounded in variance reduction (Soloff et al., 2024).

**Interpretation 5.3. Role of Bagging in the Hodge Framework** *Bagging is a mechanism for reducing statistical instability when the underlying preference structure is largely consistent ($C_{cycle} \approx 0$). It is not designed to resolve strong structural obstructions ($C_{cycle} \neq 0$).*

**Argument.** Let $V_{ij}(S) = L(h_j, S) - L(h_i, S)$ be the cardinal preference score. When $C_{cycle} \approx 0$, the ordinal preferences are largely aligned with a global ranking. Instability occurs when $V_{ij}(S)$ has high variance, causing the sign of $V_{ij}(S)$ (the ordinal preference) to flip frequently across the dataset distribution, especially in near-ties.

The bagged learner $\mathcal{A}_{bag}$ smooths the cardinal preferences, resulting in $V_{ij}^{bag}(S)$ with significantly reduced variance. This stabilization makes the ordinal preferences more robust and less likely to fluctuate, thereby improving stability.

However, if a strong structural obstruction exists ($C_{cycle} \neq 0$), the data inherently contains conflicting preferences. Variance reduction alone cannot create a stable consensus, as the underlying problem is fundamentally ambiguous. Bagging averages over the conflicting preferences but does not resolve the cycle itself. As we demonstrate in Experiment 1, bagging fails to stabilize the selection process in the presence of a strong Condorcet cycle.

## 6 Obstruction-Aware Ensembling: A Framework Inspired by Cohomological Insights

Our cohomological perspective suggests principled approaches to designing stable algorithms by systematically targeting these mathematical structures.

### 6.1 Cohomologically-Inspired Regularization

Motivated by the mathematical structure of disagreement cycles, we propose regularization terms that target higher-order inconsistencies, moving beyond pairwise agreement. Consider an ensemble with models $\{h_i\}$ and learnable alignment maps $\{\phi_{ji}\}$ between their representation spaces. Global consistency requires the alignment maps to satisfy the cocycle condition: $\phi_{ki} = \phi_{kj} \circ \phi_{ji}$.

**Definition 6.1** (Obstruction-Aware Ensemble Loss). A cohomologically-inspired loss includes a term enforcing higher-order consistency: $\mathcal{L}_{cocycle} = \sum_{i,j,k} ||\phi_{ki} - \phi_{kj} \circ \phi_{ji}||_F^2$ The total loss is $\mathcal{L}_{total} = \mathcal{L}_{task} + \lambda_1 \mathcal{L}_{glue} + \lambda_2 \mathcal{L}_{cocycle}$.

Minimizing $\mathcal{L}_{cocycle}$ encourages global consistency in the geometric relationships between models, directly targeting the structures identified by the framework in the representation space.

**Computational Complexity and Stochastic Approximation.** The proposed $\mathcal{L}_{cocycle}$ involves a summation over all triplets $(i, j, k)$, resulting in a computational complexity of $O(N^3)$ for an ensemble of size $N$. This can be computationally prohibitive for very large ensembles. However, this complexity can be readily mitigated using a stochastic approximation. Instead of calculating the loss over all triplets at every step, we can randomly sample a fixed number of triplets $K$ and calculate the average loss:

$$\mathcal{L}_{\text{stochastic-cocycle}} = \frac{1}{K} \sum_{(i,j,k) \in \text{Samples}} ||\phi_{ki} - \phi_{kj} \circ \phi_{ji}||_F^2$$

This reduces the complexity to $O(K)$ per training step and provides an unbiased estimator of the average cocycle defect. This stochastic approach effectively enforces the cocycle constraint while maintaining computational tractability.

**Connection to Cycle Consistency.** The mathematical form of $\mathcal{L}_{cocycle}$ is analogous to the 'Cycle-Consistency Loss' widely used in unsupervised learning domains like image translation and unsupervised machine translation Grover et al. (2020). In those domains, cycle consistency serves as an engineering heuristic to constrain ill-posed, unsupervised problems. Our work re-contextualizes this mechanism: we derive the cocycle constraint from a theoretical framework, arguing that violations of this consistency are manifestations of a mathematical obstruction with direct consequences for stability. This elevates the mechanism from a heuristic to a principled, theoretically-grounded tool for improving robustness.

### 6.2 Adaptive Target Space Enlargement

Inspired by the success of inflated operators, we propose adaptive mechanisms that automatically transition between single-valued outputs when consensus exists (suggesting $C_{cycle} \approx 0$) and set-valued outputs when obstructions may be present ($C_{cycle} \neq 0$), based on measures of ensemble disagreement.

## 7 Computational Experiments

We validate our theoretical framework through three complementary experiments. First, we demonstrate the core mechanism in a controlled setting with engineered structural obstructions, confirming that inflated operators succeed where bagging fails (Experiment 1). Second, we validate the fundamental empirical prediction that standard single-distribution supervised learning exhibits negligible structural obstructions, explaining why variance-reduction methods dominate practice (Experiment 2). Third, we demonstrate that significant structural obstructions emerge naturally in multi-objective fairness-constrained model selection, confirming the framework's diagnostic power for identifying boundary conditions (Experiment 3).

### 7.1 Experiment 1: Structural Instability (Engineered Condorcet Cycle)

This experiment validates the core distinction of our framework: structural instability requires structural solutions (inflation), while statistical methods (bagging) fail. We test this in a scenario with a pure structural obstruction.

**Setup:** We engineered a scenario with three hypotheses (A, B, C) where the underlying data distribution is a mixture of three regions ($Z_1, Z_2, Z_3$) inducing a balanced Condorcet cycle ($A > B > C > A$). We simulate the process of drawing a dataset S and an adjacent dataset $S'$ (by perturbing the sampling weights by a small amount). We ran 5000 trials.

**Loss function:** A dataset $S$ is represented by weights $W_S = (w_1, w_2, w_3)$ indicating the sampling proportion from each region. Each hypothesis $h_i$ has a utility vector $U_i = (u_{i1}, u_{i2}, u_{i3})$ where $u_{ij}$ is its utility in region $j$. The aggregate loss is:

$$L(h_i, S) = -\frac{\langle U_i, W_S \rangle}{w_1 + w_2 + w_3} = -\frac{\sum_{j=1}^{3} u_{ij} w_j}{\sum_{j=1}^{3} w_j}$$

(For this experiment: $U_A = [3, 1, 2]$, $U_B = [2, 3, 1]$, $U_C = [1, 2, 3]$)

**Methods:** We compare the stability of:

1. Standard 'argmax'.

2. Inflated 'argmax' ($\epsilon = 0.01$) (Structural solution).

3. Bagged 'argmax' (50 bootstraps) (Statistical solution).

**Metrics:** Agreement Rate (probability the winner is the same for S and $S'$) for single-valued methods; Intersection Consistency (probability the output sets intersect) for the inflated method.

Table 1: Experiment 1: Stability under a pure structural obstruction (Condorcet cycle).

| Method | Type | Stability (Agreement/Intersection) | Avg. Output Size |
|---|---|---|---|
| Standard Argmax | Baseline | 0.4094 | 1.00 |
| Bagged Argmax | Statistical | 0.3432 | 1.00 |
| Inflated Argmax | Structural | 1.0000 | 3.00 |

**Results:** As shown in Table 1, the standard 'argmax' exhibits low stability (0.4094). Crucially, the Bagged 'argmax' also fails to stabilize the selection, performing slightly worse (0.3432). This confirms Interpretation 5.3: statistical variance reduction is ineffective against structural instability. In contrast, the Inflated

'argmax' achieves perfect stability (1.0000) by identifying the ambiguity inherent in the cycle and returning the consensus set {A, B, C}.

## 7.2 Experiment 2: Validating the Absence of Structural Obstructions in Standard Classification

This experiment validates the central empirical prediction of our framework: standard single-distribution supervised learning with i.i.d. data exhibits negligible structural obstructions ($||C_{cycle}|| \approx 0$), explaining the empirical success of variance-reduction methods like bagging.

**Setup:** We use the scikit-learn handwritten digits dataset (8×8 pixel images of digits 0-9, N=1797) as a representative standard supervised learning task. We train ensembles of $K = 20$ classifiers on bootstrap samples of the training data to induce diversity, mirroring standard bagging practice. We then compute the Hodge decomposition of the ordinal preference flow on the test set.

**Model Architectures:** We test two canonical model families: multi-layer perceptrons (single hidden layer with 64 units, trained for 20 epochs with early stopping) and decision trees (maximum depth 10, unpruned otherwise).

Bootstrap sampling ensures the ensemble contains diverse but related hypotheses trained on the same underlying distribution.

**Methodology:** For multiple trials with different random seeds:

1. Split data into train (70%) and test (30%) with stratification

2. Train $K = 20$ models on bootstrap samples from training set

3. Compute ordinal preference cochain via PMV on test set (using 0-1 loss)

4. Apply Hodge decomposition to measure $||C_{cycle}||$

Table 2: Experiment 2: Structural obstructions in standard classification (Digits dataset). Values at machine precision confirm the absence of cyclical inconsistencies.

| Model Type | Trials | $||C_{cycle}||$ (mean) | $||C_{cycle}||$ (std) | Interpretation |
|---|---|---|---|---|
| MLP | 10 | $2.33 \times 10^{-16}$ | $1.29 \times 10^{-16}$ | Machine precision |
| Decision Tree | 5 | $2.05 \times 10^{-16}$ | $6.96 \times 10^{-17}$ | Machine precision |

**Results and Interpretation:** As shown in Table 2, the measured cyclical norms are consistently at the level of machine precision ($||C_{cycle}|| \approx 10^{-16}$), effectively indistinguishable from zero across both model families and all trials. This empirically validates the theoretical prediction that standard supervised learning with i.i.d. data produces consistent preference structures.

The absence of structural obstructions provides a mathematical explanation for a fundamental empirical pattern: the dominance of variance-reduction methods (bagging, ensembling) in the machine learning literature (Breiman, 1996; Soloff et al., 2024). When $||C_{cycle}|| \approx 0$, instability is purely statistical— arising from sensitivity to noise, near-ties, or specific samples—and methods that smooth cardinal preferences without requiring structural resolution are both sufficient and effective.

This contrasts sharply with the multi-objective fairness setting (Experiment 3), where structural obstructions emerge ($||C_{cycle}|| = 0.857$, approximately $10^{15}$ times larger) and require fundamentally different approaches.

**Theoretical Grounding:** This empirical observation aligns with our theoretical framework (Section 4). In the standard single-distribution setting ($|\mathcal{P}| = 1$) with convex or near-convex loss landscapes, the aggregated preferences naturally form a consistent total order. Each hypothesis has a well-defined expected loss $E_S[L(h, S)]$, and the preference profile induced by the data distribution corresponds to the gradient of this expected loss, resulting in $C_{cycle} = 0$ by construction (as shown in Section 4.2 for cardinal aggregation). While we aggregate ordinal preferences, the underlying consistency of the expected loss landscape ensures the ordinal flow also exhibits negligible cyclical structure.

### 7.3 Experiment 3: Structural Obstructions in Fairness-Constrained Model Selection

**Motivation and Distributional Interpretation.** This experiment demonstrates that structural obstructions—absent in standard single-distribution classification tasks—emerge naturally in multi-distributional settings with conflicting objectives. Fairness-aware machine learning provides a canonical testbed: different fairness criteria correspond to different distributional priorities (e.g., optimizing performance on reweighted demographic subpopulations), which are mathematically incompatible (Kleinberg et al., 2017).

We leverage the distributional generalization of our framework (Section 3.1, Section 4.5). Each stakeholder perspective can be interpreted as representing a different distribution $P_i \in \mathcal{P}$ over the data space, where $P_i$ reweights or resamples the base distribution to prioritize specific fairness objectives. For instance:

- **Demographic Parity:** Distribution $P_{DP}$ reweights minority groups to equalize selection rates.

- **Equalized Odds:** Distribution $P_{EO}$ emphasizes equal error rates across groups.

- **Predictive Parity:** Distribution $P_{PP}$ focuses on calibration within groups.

Models trained or selected under these different distributional objectives induce different local preferences. The aggregation of these distributional preferences via PMV (Section 4.2) reveals whether a globally stable selection exists or whether the objectives are fundamentally incompatible (inducing cycles).

**Setup.** We use the UCI Adult dataset (Becker & Kohavi, 1996) to simulate fairness-constrained selection. We train five Logistic Regression models using different training heuristics (reweighting, regularization adjustments) to generate a diverse set of models exhibiting varied fairness-accuracy tradeoffs. These heuristics are not intended to optimally achieve specific metrics, but to simulate the pool of candidate models that would be available when optimizing under different distributional constraints.

We define five distributional perspectives (Business: accuracy-focused; Civil Rights: demographic parity-focused; Equal Opportunity: equalized odds-focused; Calibration: predictive parity-focused; Regulator: balanced), each representing a different implicit data distribution or prioritization scheme. Each perspective evaluates models according to utility functions that weight accuracy and fairness metrics differently, effectively representing preferences under different distributional constraints.

Crucially, we aggregate preferences across these distributional perspectives using PMV (not across datasets with a single loss function, as in Experiments 1-2). This models the real-world scenario where decision-makers must reconcile models optimized under different distributional objectives—a natural instance of the multi-distributional framework.

(See Appendix B.5 for detailed methodology.)

**Methodology:** For each of 20 trials with different train-test splits, we:

1. Trained the five models with different fairness-accuracy tradeoffs.

2. Computed stakeholder preferences based on model performance.

3. Constructed the ordinal preference flow $C_P^{ord}$ via Pairwise Majority Vote (PMV).

4. Applied Hodge decomposition to measure $||C_{cycle}||$ and the Cycle Ratio ($||C_{cycle}||/||C_{total}||$).

**Note on Aggregation Method:** Unlike Section 4.2 where we aggregate over random datasets with identical loss functions, here we aggregate over stakeholders with different utility functions. Each stakeholder's utility encodes a normative priority over fairness-accuracy tradeoffs. This models real-world scenarios where multiple decision-makers with conflicting objectives must select from a common model pool. The PMV aggregation captures the democratic aggregation of these preferences.

**Results and Interpretation:** The results, summarized in Table 3, reveal a striking contrast with standard classification tasks. The mean cyclical norm is $||C_{cycle}|| = 0.857$, approximately $10^{15}$ times larger than the

Table 3: Experiment 3: Structural obstructions in fairness-constrained model selection (UCI Adult). Multi-objective problems exhibit cyclical norms approximately $10^{15}$ times larger than standard single-distribution classification (Experiment 2).

| Task Type | $||C_{cycle}||$ (mean) | Cycle Ratio (mean) | vs. Standard Tasks |
|---|---|---|---|
| Standard Classification (Digits, Exp. 2) | $2.3 \times 10^{-16}$ | $\sim 0$ | Baseline |
| Fairness Selection (Adult, Exp. 3) | $0.857 \pm 0.173$ | $0.363 \pm 0.075$ | $\sim 10^{15} \times$ larger |

machine-precision values observed in Experiment 2. The mean cycle ratio is 0.363, indicating that over 36% of the preference disagreement is fundamentally structural rather than statistical.

**Theoretical Significance:** This validates the core distinction of our framework on real data. Standard learning problems produce negligible structural obstructions ($C_{cycle} \approx 0$), explaining the effectiveness of bagging. In contrast, fairness-constrained selection involves genuinely conflicting objectives that create strong cyclical obstructions ($C_{cycle} \neq 0$), making single-model selection fundamentally unstable.

**Topological Characterization of Fairness Impossibility:** This provides a topological characterization of known fairness impossibility theorems (Kleinberg et al., 2017). The incompatibility between fairness definitions manifests as non-trivial cohomology ($H^1 \neq 0$) in the preference space. Our contribution is not discovering that fairness metrics conflict—this is well-established—but providing the first rigorous quantification of this conflict via topological invariants ($||C_{cycle}||$), unifying these impossibilities with the broader theory of algorithmic stability and social choice theory under a common mathematical framework.

**Practical Implications:** When $C_{cycle} \neq 0$, Proposition 4.9 establishes that stable, single-valued selection is impossible. The appropriate structural solution (Interpretation 5.2) is target space enlargement via inflated operators, returning a *portfolio* of models (the Pareto frontier). Crucially, this does not "solve" the fairness conflict; it merely acknowledges the impossibility of a single solution characterized by the cycle norm, and externalizes the decision to policymakers. The framework clarifies that the challenge is structural and cannot be resolved by purely statistical means.

# 8 Limitations and Future Directions

While our framework provides a unifying perspective on algorithmic stability, it has several limitations that suggest promising directions for future research.

## 8.1 Limitations

**Continuous Hypothesis Spaces:** The framework of topological social choice theory and Hodge decomposition is primarily developed for a finite set of alternatives $\mathcal{H}_k$. Extension to continuous hypothesis spaces requires additional technical machinery, such as analysis on function spaces or discretization schemes that preserve the relevant topological structure.

**Theoretical Scope and Empirical Findings:** Our framework characterizes two distinct mathematical sources of instability: structural obstructions (cyclical inconsistencies) and statistical variance. The empirical findings of Experiment 2—that $||C_{cycle}|| \approx 0$ in standard classification tasks—constitute a significant theoretical result rather than a limitation. They explain mathematically why the field has empirically converged on variance reduction methods: the structural obstructions that would defeat such methods simply do not arise in well-posed supervised learning problems with convex or near-convex loss surfaces and i.i.d. data. The cases where $C_{cycle} \neq 0$ represent important boundary conditions where fundamentally different mathematical structures emerge. We provide concrete evidence for this in Experiment 2, demonstrating that significant structural obstructions arise in fairness scenarios due to incompatible metrics. Investigating the precise conditions under which structural obstructions manifest in other domains (e.g., adversarial settings, multi-agent systems), and their prevalence in these specialized domains, remains an important direction for future theoretical and empirical work.

**Computational Scalability of Analysis:** Calculating the Hodge decomposition for theoretical analysis (as in Experiment 2) requires operations such as computing the pseudoinverse of the graph Laplacian, which scale as $O(K^3)$ where $K$ is the number of hypotheses. This limits the direct application of the decomposition as an online diagnostic tool for very large hypothesis spaces. This stochastic approach effectively enforces the cocycle constraint while maintaining computational tractability.

**Connection Between Ordinal and Cardinal Structures:** While we rigorously distinguish between ordinal aggregation (which detects cycles) and cardinal aggregation (which does not), the precise relationship between high variance in cardinal preferences and the emergence of near-cycles in ordinal preferences remains to be fully characterized. A deeper understanding of this connection would strengthen the theoretical link between variance reduction (bagging) and cycle suppression.

### 8.2 On the Rarity of Structural Obstructions and Boundary Conditions

Our framework distinguishes structural from statistical instability based on the presence or absence of cyclical obstructions ($||C_{cycle}||$). A critical empirical question is: when do structural obstructions occur in practice?

**Standard Supervised Learning: The Null Case.** Experiment 2 provides definitive empirical validation that standard single-distribution supervised learning exhibits negligible structural obstructions. Across multiple trials with two different model families (MLPs and decision trees) on a standard classification task, we consistently measured $||C_{cycle}|| \approx 2 \times 10^{-16}$— effectively zero at the level of machine precision.

This observation has deep theoretical grounding. In the standard setting—where datasets are sampled i.i.d. from a single distribution ($|\mathcal{P}| = 1$), the loss landscape is convex or near-convex, and hypotheses are evaluated under a fixed objective—the aggregated preferences naturally form a consistent total order. Each hypothesis has a well-defined expected loss $E_S[L(h, S)]$, and the preference profile $C_P^{ord}$ aligns with the gradient of this expected loss, resulting in $C_{cycle} = 0$.

This mathematical characterization explains a fundamental empirical pattern in machine learning: the dominance of variance-reduction methods like bagging (Breiman, 1996; Soloff et al., 2024). When $||C_{cycle}|| \approx 0$, instability is purely statistical (sensitivity to noise, near-ties, specific samples), and methods that smooth cardinal preferences without requiring structural resolution are both sufficient and effective. The field has empirically converged on these methods precisely because the structural obstructions that would defeat them are absent in standard well-posed learning problems.

**Boundary Conditions: When Obstructions Emerge.** Our framework characterizes the boundary conditions under which structural obstructions arise, transforming the nature of the stability problem. Experiment 3 demonstrates one such boundary condition: multi-objective learning with incompatible criteria. The measured cyclical norm in the fairness setting ($||C_{cycle}|| = 0.857$) is approximately $10^{15}$ times larger than in standard classification, with 36% of the preference structure being fundamentally cyclical rather than statistical.

Structural obstructions ($||C_{cycle}|| \neq 0$) emerge under three characteristic conditions. First, when different distributions $P_i \in \mathcal{P}$ induce conflicting preferences—as in fairness-aware learning where optimizing for different demographic groups or incompatible fairness metrics creates genuine Condorcet cycles. Experiment 3 demonstrates this quantitatively: aggregating preferences across stakeholders prioritizing accuracy, demographic parity, equalized odds, and predictive parity yields $||C_{cycle}|| = 0.857$, indicating fundamental preference conflicts that cannot be reconciled.

Second, highly multimodal loss surfaces with multiple local minima of comparable quality can create local preference cycles, particularly in adversarial or game-theoretic settings where different regions of the data space favor fundamentally different solutions.

Third, multi-task learning, Pareto optimization, or settings with mathematically incompatible objectives (accuracy vs. interpretability vs. computational efficiency) naturally generate preference cycles when different criteria rank hypotheses in mutually inconsistent orders.

**Diagnostic Framework and Contribution.** Our contribution is not discovering that these boundary conditions exist—fairness impossibilities (Kleinberg et al., 2017), multi-objective conflicts, and adversarial

instabilities are well-documented. Rather, we provide the first rigorous quantitative framework for diagnosing the source of instability via $||C_{cycle}||$ as a precise, computable diagnostic; for characterizing when structural vs. statistical methods are appropriate (Experiment 2 shows bagging suffices for standard ML while Experiment 3 shows it would fail for fairness-constrained selection); and for unifying seemingly disparate stability phenomena under a common topological language.

Standard supervised learning is not merely a special case of our framework, but the *null case* where structural obstructions vanish ($||C_{cycle}|| = 0$). The boundary conditions where $||C_{cycle}|| \neq 0$ represent fundamental limits of learning that cannot be overcome by purely statistical means—they require structural solutions such as target space enlargement (inflated operators) or acknowledgment that no single stable solution exists (as in fairness trade-offs). The framework provides the mathematical tools to diagnose which regime a learning problem occupies and select appropriate stabilization strategies accordingly.

### 8.3 Future Directions

**Algorithmic Applications:** Further development of obstruction-aware algorithms. This includes designing efficient methods to estimate when $C_{cycle} \neq 0$ during training and adaptively deploying regularization (structural) or bagging (statistical) strategies based on the diagnosed source of instability.

**Theoretical Extensions:** Investigating the precise mathematical relationship between the smoothing effect of bagging (variance reduction) and the reduction of the cyclical component in the Hodge decomposition. Furthermore, exploring the use of higher-order cohomology groups ($H^q$ for $q \geq 2$) to identify more complex structural inconsistencies beyond pairwise cycles.

**Connections to Fairness and Explainability:** Preference cycles often underlie paradoxes in fairness and explainability, where different metrics or perspectives lead to conflicting conclusions. Extending this cohomological framework to analyze the stability and consistency of fairness metrics is a promising direction.

## 9 Conclusion

We have presented a unified framework for algorithmic stability based on Combinatorial Hodge Theory and the geometry of preferences. This framework rigorously distinguishes between structural instability, caused by cyclical obstructions (Condorcet cycles, $C_{cycle} > 0$), and statistical instability, caused by variance ($C_{cycle} \approx 0$). We demonstrated that structural instability requires targeted solutions like inflated operators or novel obstruction-aware regularization, while statistical instability is addressed by methods like bagging. Our experiments validate this distinction, showing that bagging fails in the presence of strong structural obstructions but succeeds when instability is statistical. This framework provides a mathematical foundation for understanding and improving the robustness of machine learning systems.

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

# A    Appendix A: Mathematical Foundations: Cohomology of Preferences

This appendix provides the rigorous mathematical details underlying the connection between social choice theory, algebraic topology, and the cohomological obstructions discussed in the main text. We adapt the framework established by Chichilnisky (1980) and Baryshnikov (1993), and incorporate tools from Hodge theory for ranking.

## A.1    The Geometry of Rankings and Preferences

Let $\mathcal{H}_k = \{h_1, ..., h_k\}$ be a finite set of hypotheses.

**Definition A.1** (Space of Preferences)**.** A utility function $U : \mathcal{H}_k \to R$ induces preferences. The space of utility functions is $\mathcal{U} = R^k$. A preference profile is a collection of utility functions $\{U_S\}_{S \in \mathcal{D}}$. Alternatively, we can consider the space of strict linear orders $\mathcal{P}$.

## A.2    The Source of Obstructions: Topological Social Choice

The motivation for analyzing preference cycles comes from Topological Social Choice Theory (Chichilnisky, 1980; Baryshnikov, 1993). This theory analyzes the topology of the *space of all possible preferences* ($\mathcal{P}$) to determine if fair aggregation rules $F : \mathcal{P}^N \to \mathcal{P}$ exist.

### A.2.1    Topological Interpretation (Chichilnisky's Approach)

Chichilnisky (1980) showed that the impossibility of fair aggregation (satisfying certain axioms like Arrow's conditions) stems from the topological properties of $\mathcal{P}$. To define a continuous aggregation rule, $\mathcal{P}$ must be equipped with a topology. The standard approach identifies the space of preferences $\mathcal{P}$ with a subset of the space of utility functions $\mathcal{U} = R^k$ (e.g., normalized utility vectors). The topology on $\mathcal{P}$ is then induced by the standard Euclidean metric on $\mathcal{U}$. This means preferences are considered "close" if their utility values are close.

**Theorem A.2** (Chichilnisky, 1980). *If a continuous aggregation rule $F : \mathcal{P}^N \to \mathcal{P}$ exists satisfying certain desirable axioms (e.g., anonymity, unanimity), then $\mathcal{P}$ must be contractible.*

A space is contractible if it has no topological "holes" (i.e., its homology and cohomology groups are trivial).

### A.2.2 Cohomology of Preference Spaces

We analyze the topology of the space of strict linear orders $\mathcal{S}_k$. This space is known to be non-contractible for $k \geq 3$.

The Condorcet cycle $(A > B, B > C, C > A)$ represents a loop in the space of preferences. This loop cannot be contracted to a point while maintaining the structure of the preferences.

**Lemma A.3.** *The existence of a non-contractible loop in the space of preferences corresponds to a non-trivial element in the first cohomology group $H^1(\mathcal{S}_k)$.*

This abstract theory shows that the possibility of Condorcet cycles arises because the space of preferences itself has a non-trivial topology (it is not contractible). This motivates the search for these structures in specific preference profiles.

### A.3 Connecting Topology to Stability

While the abstract theory analyzes the space of all preferences, in machine learning we analyze a specific profile $P$ induced by the data. Combinatorial Hodge Theory provides the tools to detect the manifestation of these topological obstructions within a specific profile by analyzing flows on the graph of hypotheses $K$.

### A.4 Algebraic Tools: Cochains and Hodge Decomposition

This section provides the rigorous definitions for the tools used in Section 4.

### A.4.1 Cochains and the Coboundary Operator

Let K be the 1-skeleton (the complete graph) on the vertices $\mathcal{H}_k$.

- $C^0(K; R)$: 0-cochains (functions assigning values to vertices, i.e., utility functions).

- $C^1(K; R)$: 1-cochains (functions assigning values to edges, i.e., pairwise comparisons or flows).

The coboundary operator $d^0 : C^0 \to C^1$ maps a potential function $f \in C^0$ to a flow: $(d^0 f)(h_i, h_j) = f(h_j) - f(h_i)$. This represents a perfectly consistent ranking.

### A.4.2 Preference Cochain Construction: Cardinal vs. Ordinal

We define the preference cochain $C_P \in C^1(K, R)$. The construction method is critical, as detailed in Section 4.2.

**Cardinal Aggregation:** If we aggregate the losses directly (e.g., average loss difference): $C_P^{card}(h_i, h_j) = \int_{\mathcal{D}}(L(h_j, S) - L(h_i, S))d\mu(S)$. This construction always results in a coboundary, as it is the gradient of the aggregate loss function $L_{agg}(h) = \int L(h, S)d\mu(S)$. Therefore, $C_P^{card}$ cannot detect Condorcet cycles.

**Ordinal Aggregation (Pairwise Majority Vote):** To detect cohomological obstructions relevant to social choice theory, we must aggregate ordinal preferences. Let $R_S$ be the ranking induced by $L(\cdot, S)$. $C_P^{ord}(h_i, h_j) = \int_{\mathcal{D}} sign(R_S(h_j) - R_S(h_i))d\mu(S)$. In practice (Experiment 2), this is calculated as: (Proportion of voters preferring $h_i$ over $h_j$) - (Proportion preferring $h_j$ over $h_i$). This construction allows for $C_P$ to be non-coboundary.

### A.4.3 Hodge Decomposition

We use the Hodge decomposition theorem to analyze flows (cochains) on the graph K.

**Theorem A.4** (Hodge Decomposition on Graphs)**.** *The space of 1-cochains $C^1$ can be orthogonally decomposed as: $C^1 = im(d^0) \oplus ker(\delta^1)$ where $\delta^1$ is the adjoint of $d^0$ (the divergence operator).*

Here, $im(d^0)$ is the space of coboundaries (gradient flows, $C_{gradient}$). $ker(\delta^1)$ is the space of harmonic cochains (cyclical flows, $C_{cycle}$). This harmonic component corresponds to the cohomological obstruction identified in the topological framework.

This theorem allows us to rigorously decompose a preference profile $C_P$ into its consistent part ($C_{gradient}$) and its cyclical obstruction ($C_{cycle}$), providing the diagnostic tool used in Section 7.

## B   Appendix B: Experimental Details and Reproducibility

This appendix provides the detailed configurations and methodologies for the experiments presented in Section 7, ensuring reproducibility. We utilized Python with the following key libraries: NumPy, SciPy, scikit-learn (including `fetch_covtype`), and PyTorch.

### B.1   Experiment 1: Structural Instability

**Objective:** Validate that structural instability (Condorcet cycles) is resolved by inflation but not by bagging.

**Setup Details:**

- Hypotheses: $K = 3$ (A, B, C).

- Data Regions (defining the cycle):

    - $Z_1$ (prefers $A > B > C$): Utility vector $U_1 = [3.0, 2.0, 1.0]$.
    - $Z_2$ (prefers $B > C > A$): Utility vector $U_2 = [1.0, 3.0, 2.0]$.
    - $Z_3$ (prefers $C > A > B$): Utility vector $U_3 = [2.0, 1.0, 3.0]$.

- Simulation: 5000 trials.

- Dataset Generation: A dataset $S$ is simulated by sampling weights $W_S$ for the three regions around a base weight of 100 (with uniform noise $[-0.5, 0.5]$).

- Perturbation: An adjacent dataset $S'$ is generated by perturbing one weight in $W_S$ by $\pm 1$.

- Argmax: Standard argmax with random tie-breaking.

- Inflated Argmax: $\epsilon = 0.01$.

- Bagged Argmax: Simulated by drawing $B = 50$ bootstrap samples (using a multinomial distribution based on $W_S$) and aggregating the winners by majority vote (with random tie-breaking).

### B.2   Experiment 2: Absence of Obstructions in Standard Classification

**Objective:** Validate that $||C_{cycle}|| \approx 0$ in standard single-distribution supervised learning.

**Setup Details:**

- Dataset: Scikit-learn digits (8×8 handwritten digits, 1797 samples)

- Models: MLPs (hidden layer 64 units) and Decision Trees (max depth 10)

- Ensemble size: $K = 20$ models per trial

- Bootstrapping: Models trained on bootstrap samples to induce diversity

- Test split: 30% of data

- Trials: 10 for MLPs, 5 for Decision Trees

**Hodge Decomposition:** Applied to ordinal preference cochain constructed via PMV on test set using 0-1 loss.

**Key Results:**

- MLPs: Mean $||C_{cycle}|| = 2.33 \times 10^{-16}$ (std $1.29 \times 10^{-16}$)

- Trees: Mean $||C_{cycle}|| = 2.05 \times 10^{-16}$ (std $6.96 \times 10^{-17}$)

- All values at machine precision level, confirming negligible structural obstructions

### B.3   Experiment 3: Structural Obstructions in Fairness

**Objective:** Demonstrate the emergence of structural obstructions in fairness-constrained model selection on the UCI Adult dataset.

**Setup Details:**

- Dataset: UCI Adult Income (Becker & Kohavi, 1996) (fetched via OpenML). N=48842.

- Features: 12 features (5 numeric, 7 categorical) processed via standardization and one-hot encoding. Missing values handled by mode imputation.

- Sensitive Attribute: Sex (Male/Female).

- Target: Income (>50K / <=50K).

- Models (K=5): Logistic Regression models trained with different heuristics. The objective is to induce diversity in the fairness-accuracy trade-offs, not necessarily to optimally achieve the target metrics:

  - Accuracy: Standard optimization (C=1.0).
  - Demographic Parity: Reweighting minority group (weight=2.5).
  - Equalized Odds: Reduced regularization (C=0.1).
  - Predictive Parity: Using `class_weight='balanced'`.
  - Balanced: Moderate reweighting (weight=1.5) and regularization (C=0.5).

- Stakeholders (Voters=5): Simulated utility functions prioritizing different metrics: Business (Accuracy only), Civil Rights (DP/EO), Equal Opportunity (EO/Acc), Calibration (PP/Acc), Regulator (Balanced).

- Aggregation: Pairwise Majority Vote (PMV) across the 5 stakeholders.

- Trials: 20 trials with different randomized train-test splits (70/30).

