# OpenReview forum: "The Geometry of Algorithmic Stability: A Hodge Theoretic View on Structural vs. Statistical Instability"
_TMLR — Accepted by TMLR_

### Review · Reviewer_EcaC · 2025-09-22

**Summary Of Contributions:**

The authors revisit algorithmic stability from a topological perspective. A classification algorithm (mapping datasets S to hypotheses
h) is stable if similar datasets are classified the same way.

- They analyze this property in terms of the absence of cycles in a graph over hypotheses induced by pairwise (ordinal) preferences.
- They then argue that well-known methods (e.g., bagging or output-space enlargement) can be viewed as reducing or eliminating such cycles.
- Finally, they propose a “cohomologically inspired” regularization that discourages these cycles when minimizing the training loss.

**Audience:**

Yes

**Audience Explanation:**

The overall idea is fun and appealing. I especially like the attempt to view several methods that seem unrelated at first (e.g., bagging and output-space enlargement) as actually trying to achieve the goal.

One criticism: at times the paper leans heavily on cohomology jargon, but the concrete machinery used is the Hodge decomposition of a flow on a graph. Therefore, I’d minimize the topological vocabulary (you likely only need graph theory here) to make the paper more accessible to the average ML researcher (the primary audience for TMLR).

**Broader Impact Concerns:**

No particular concerns.

**Claims And Evidence:**

No

**Claims Explanation:**

My main issue is that it’s hard to make sense of the mathematical results as stated: I couldn’t connect the definitions to the proofs. Since the paper proposes a new framework, that framework needs to be well defined and clearly presented.



___________________________


**Example 1**

Theorem 4.2: "*A preference profile P contains a Condorcet cycle if and only if the configuration of local preferences leads to a structural inconsistency. This inconsistency is mathematically characterized by the existence of a non-trivial class in the first cohomology group H1 of the underlying topological space of preferences.*"

- It is never explained what it means for a "preference profile P to contain a Condorcet cycle". An example of Condorcet cycle is given somewhere: "A>B, B>C,C>A", but what is the preference profile there? I have no idea what this means.
- What does it means that there is "a non-trivial class in the first cohomology group H1 of the underlying topological space of preferences"? For this to make sense, there should be topology on the space of preferences. Where is this topology defined?

The proof points to Appendix A, but that didn’t resolve these gaps for me.

__________

**Example 2:**

Proposition 4.4. *If the preference profile induced by the data contains a structural inconsistency (a Condorcet cycle characterized by a non-trivial H1 class), then for sufficiently small accuracy tolerance δ and stability parameter ε, the Stable Selection Problem is unsolvable.*

I still don’t know what the assumption precisely means. Reading the proof, it seems to rely on a stronger condition: there are disjoint regions of datasets where different hypotheses are preferred with a uniform margin (which is not what the statement says I believe). Maybe that stronger condition is true, maybe not. As written, the definition isn’t clear, and there’s a statement–proof mismatch.


__________

**Example 3:**

Proposition 5.1. *Bagging transforms the original preference profile P induced by an unstable base learner A into a smoother profile Pbag induced by the bagged learner Abag. This operation systematically reduces the magnitude of the cohomological obstruction in the aggregated ordinal preferences.*

In the proof, the paper shifts to a different (and much clearer) object: a preference cochain C_P^ord (= function on edges), which admits a standard Hodge decomposition into a gradient part plus a cyclical part (C_P^ord)_{cycle}.  The proof is saying that on expectation, after bagging,

$$
|C^{P_{bag}}_{ord,cycle}| \leq |C^{P_{bag}}_{ord}|,  (*)
$$

ie "bagging reduces the number of cycles". That’s a precise mathematical target! But then:
- Should earlier claims (Theorem 4.2, Prop. 4.4) also be framed directly in terms of C^P_{ord} and its Hodge split? If so, say so up front. This seems central but is only detailed in the appendix.
- The argument for (∗) reads as a heuristic, not a proof. Heuristics are fine (this is not a math journal), but it should be clearly marked as such.
______

Summary: The paper’s big picture is interesting, but the claims aren’t backed by clear, precise definitions and proofs. At minimum, the paper should (i) define what is the topology of the preference profile space or switch consistently to the graph/Hodge setting with  C^P_ord, (ii) align statements with the assumptions actually used in proofs, and (iii) clearly distinguish formal results from heuristics.
	​

**Requested Changes:**

Requested changes were given in the two previous sections.

Additional comments:
- What is the dataset space D? It is said that "D is the set of finite training datasets", which is seen "as a graph where edges connect adjacent datasets (differing by one point)", and then later that we assume that "the dataset space D is connected." But the set of all finite subsets of a given set is clearly connected right? I am confused on what D should be.

---

> ### Author Response · Authors · 2025-10-02
>
> We thank the reviewer for the engagement with the mathematical framework and the insightful suggestions for improvement.
>
> **W1/E1: Undefined Mathematical Foundations (Topology of Preference Space, Definition of Cycle).**
> The reviewer correctly pointed out critical gaps in the mathematical definitions.
> We have addressed this by implementing the reviewer's suggestion (W2) to pivot the presentation to the concrete machinery of **Combinatorial Hodge Theory** (Section 4).
>
> *   **Definition of Cycle:** We now define structural inconsistencies concretely as the cyclical component ($C_{cycle}$) in the Hodge decomposition of the Ordinal Preference Flow (Def 4.11).
> *   **Mathematical Grounding:** Section 4 provides rigorous definitions for the graph, flows, and the coboundary operator.
> *   **Topology Definition:** While the main analysis now relies on Hodge theory, we have added the definition of the topology on $\mathcal{P}$ in Appendix A.2.1 for completeness regarding the abstract motivation (Chichilnisky's Theorem).
>
> **W1/E2: Statement-Proof Mismatch (Proposition 4.4).**
> The reviewer noted that the proof of Prop 4.4 (now 4.13) relied on stronger conditions (uniform margins over disjoint regions) than stated, and only addressed the discrete metric case. We have resolved this mismatch by introducing the rigorous definition of "Strong Structural Inconsistency" (Def 4.12) and updating the proposition statement to rely on this definition. Furthermore, the proof itself has been updated to rigorously handle the general metric space case using the triangle inequality.
>
> **W1/E3: Heuristic Proofs and Clarity (Proposition 5.1).**
> The reviewer noted the argument for Prop 5.1 (regarding bagging) was heuristic and suggested framing the claims directly in terms of the Hodge split.
> We have replaced the heuristic argument with a refined interpretation (Interpretation 5.4) that rigorously distinguishes between bagging's role in reducing statistical variance vs. resolving structural obstructions. This interpretation is now strongly supported by revised experiments: Exp 1 shows bagging fails against structural instability, while Exp 2 shows it succeeds against statistical instability (where $C_{cycle} \approx 0$).
>
> **W2: Presentation Strategy (Minimize Topological Vocabulary).**
> We appreciate this suggestion and have implemented it. The paper now focuses primarily on the concrete and accessible machinery of Combinatorial Hodge Theory (flows on a graph) in Section 4, minimizing the reliance on abstract cohomology jargon.
>
> **RC: Definition of Dataset Space $\mathcal{D}$.**
> We clarified the definition of $\mathcal{D}$ in Section 3.1 (set of datasets of fixed size N, adjacency defined by differing by one point) and clarified why the connectivity assumption (Assumption 3.5) is necessary for the analysis.

---

> > ### Comment · Reviewer_EcaC · 2025-11-13
> > **Response**
> >
> > I would like to thank the authors for improving the clarity of the paper. However, I still find that the paper suffers from insufficient definitions and missing formalism, which make several parts difficult to understand. Some key objects are never formally introduced (for example the distance on $ \mathcal{P}(\mathcal{H}) $, the loss $ \mathcal{L}_{\text{glue}} $ and the datasets and models used in Experiments 3 and 4). As a result, some theoretical arguments are very difficult to follow, and in some experiments I could not reconstruct how the setup was implemented. There are also several notation and clarity issues, which further obscure the presentation.
> >
> > ## Questions
> >
> > - How is cyclical obstruction related to strong structural inconsistency? Are there any implications between the two notions? Note that the former depends on a distribution on $ \mathcal{D} $, whereas the latter is purely deterministic. It seems that the paper sometimes conflates the two notions (for example in Section 5.1 and Proposition 5.2).
> >
> > - I am very confused by Section 5.1.1. First, what is the distance used over $ \mathcal{P}(\mathcal{H}) $? Then, in the statement of Proposition 5.2, we refer to cyclical obstruction, but in the proof we refer to strong structural inconsistency. In the proof, we also use that the "cycle is balanced", but this term is never defined. Moreover, the assumption that the hypotheses not in the cycle $ H^* $ have strictly higher loss than the ones in $ H^* $ should be stated in the proposition, not introduced in the proof. More details should be added on the stability of the set-valued output. I do not understand the argument right now.
> >
> > - I do not understand what $ \mathcal{L}_{\text{glue}} $ is. What is $ x $? What is $ h_i(x) $? In Section 6.1, it would greatly help to properly introduce the statistical setting: dataset, representation spaces, task loss, and so on.
> >
> > ## Minor remarks
> >
> > - I remain unsure whether $ \mathcal{D} $ should be any subset of $ \mathcal{X}^N $ (with connectivity assumed) rather than the full set $ \mathcal{X}^N $. If $ \mathcal{X} = \mathbb{R}^K $ and $ \mathcal{H} $ is finite, it seems strange to assume that for any dataset $ S \in \mathcal{X}^N $ we have the same classification, which stability would imply. Taking $ \mathcal{D} $ as a proper subset might be more reasonable. In this case, the expectation in Section 4.2 should be with respect to a prescribed distribution on $ \mathcal{D} $.
> >
> > - Sometimes S and sometimes $ S $ is used to denote a dataset.
> >
> > - Section 4.3: What is the definition of the norm of a chain?
> >
> > - The assumption that "for sufficiently small $ \delta $, the accuracy constraint implies localization" should appear in the statement of Proposition 4.8, not only in the proof.
> >
> > - Section 5.1: The title should be $ C_{\text{cycle}} \neq 0 $, not $ > $.
> >
> > ## Experiments
> >
> > - **Experiment 1**: The setup should be more clearly explained. We do not have datasets in the traditional sense $ S = (X_1, \dots, X_n) $. Instead, a dataset is represented by a vector $ W = (W_1, W_2, W_3) $, where $ W_i $ is the number of observations in region $ Z_i $, and then
> >   $ L(h_i, S) = \langle U_i, W \rangle / (W_1 + W_2 + W_3) $.
> >   If this is the intended loss, it should be stated explicitly.
> >
> > - **Experiment 2**: Please describe the Forest Covertype dataset in a couple of sentences, including the role of the Wilderness Area covariate.
> >
> > - **Experiments 3 and 4**: I cannot reconstruct what is being done here. What is the dataset? What are the models? What is the loss being minimized? More explanation is needed.
> >
> > - **Experiment 5**: The preference chain is computed in a different way from Section 4.2. In Section 4.2 we consider the probability of $ h_i $ being preferred over $ h_j $ across different random datasets $ S $. In Experiment 5 the aggregation is instead taken over stakeholders with different utility functions. This should be explained explicitly, since it changes the interpretation of $ C_{\text{cycle}} $.

---

> > > ### Author Response · Authors · 2025-11-16
> > > **revision done**
> > >
> > > We have revised the manuscript and hope that the questions raised have been addressed systematically. Unfortunately, the overall length of the manuscript has not decreased. We are not constrained by any deadlines on our side (though TMLR may impose some), and we generally prioritize fair evaluation over a fast response.

---

### Review · Reviewer_tpE2 · 2025-09-28

**Summary Of Contributions:**

The paper draws interesting connections between topological social choice theory and algorithmic stability. It considers a finite collection of datasets, $\mathcal{D}$, viewed as a graph where two datasets are adjacent if they differ by only one data point. A learning algorithm is a map $\mathcal{D} \to \mathcal{H}$. From the point of view of social choice theory, we can view each dataset in $\mathcal{D}$ as a voter, each having some preferences over the models in $\mathcal{H}$ induced by a loss function. The learning algorithm selects the winning model given each dataset based on those preferences.

Topological social choice theory asserts that attempts to derive a global ranking of $\mathcal{H}$ based on local preferences usually lead to Condorcet cycles. The paper observes that these Condorcet cycles are a significant source of algorithmic instability: the learning algorithm does not choose "similar" hypotheses under small perturbations in the data (when datasets differ by only one data point).

The paper then analyzes how methods such as bagging and regularization prevent Condorcet cycles by smoothing the preference landscape, while inflated operators resolve Condorcet cycles by enlarging the target space. It also proposes other methods for reducing these obstructions.

**Strengths**
* The connection made between two seemingly very different fields is highly intriguing. It strengthens our existing understanding of algorithmic robustness by showing that inconsistent global rankings, which are somewhat inevitable, directly lead to algorithmic instability. This is a very novel perspective to me, and I believe it will be of interest to the ML community.

**Weaknesses**
* My primary concern with this paper lies in the writing. I believe additional effort is required to make the presentation clearer.
	* Since the mathematical framework heavily builds on topological social choice theory, I found the current Appendix 1 too brief and unclear to provide the necessary mathematical background. Furthermore, some section titles do not accurately reflect their content, and the overall logical flow of the paper feels disjointed.  (See "Requested Changes" for more specific feedback.)
* The "Propositions" and "Proofs" in this paper feel more like informal mathematical explanations rather than rigorous statements and proofs that clarify the key assumptions being made. While this level of rigor might be acceptable to some extent, it would still be better if the authors could formalize these more rigorously. If the authors instead have reasons against this, they might wish to refrain from labeling them as "Propositions/Proofs," or they could explicitly add comments in the paper clarifying this lack of formality.

**Audience:**

Yes

**Audience Explanation:**

Yes, I believe the findings of the paper make a valuable contribution to enhancing our understanding of algorithmic robustness and would be of interest to TMLR's audience.

**Broader Impact Concerns:**

This work has no ethical implications.

**Claims And Evidence:**

Yes

**Claims Explanation:**

Yes. The proofs are given for the propositions, and experiment results support the arguments of the paper.

**Requested Changes:**

These comments suggest optional changes that I believe could make the paper more readable, though the authors may disagree. The comments also highlight a few confusions I had while reading the paper --- please consider making changes where my confusions seem reasonable or might be shared by other readers.

**Introduction**
*(Optional)* I believe it would improve clarity to add a short, intuitive overview in the introduction explaining the connection between social choice theory and algorithmic stability, given that they initially seem like very different areas. For example, you could include a paragraph explaining how "datasets" are viewed as voters, and how social choice theory reveals the inevitability of the "Condorcet Paradox," which directly leads to algorithmic instability.

**Related Work**
Is there a specific reason why topological social choice theory is not addressed as its own subsection in the "Related Work" section? It seems like a good opportunity to introduce the main ideas and key conclusions of this field—especially those most relevant to the paper.

**Section 3**
* The titles of the sections and subsections in Section 3 do not align well with the content. It takes reading much of the later sections to understand what these terms mean, making the organization feel confusing.
    * For example, Section 3 is titled "Stability as Constrained Preference Aggregation." This is confusing since the term "preference aggregation" isn't fully explained until the end of Section 4.1 and Appendix A.2. Moreover, it is not clear why or how "stability" corresponds to "constrained preference aggregation," as the title suggests.
    * Similarly, Section 3.2 is titled "Accuracy-Stability Trade-Off," but the section content does not actually discuss any trade-offs. Instead, it defines "The Stable Selection Problem" as one that has both accuracy and stability. The "trade-off" seems to emerge later in Proposition 4.4, but the current organization makes this connection unclear.

**Section 4**
* **(Confusion)** Line 2 in Section 4: What does "stable aggregation" mean? Is "algorithmic stability" considered a type of "aggregation"? If so, how?
* **Terminologies**: To clarify, are the following interpretations correct?
    - "Preference" refers to an ordering on the hypotheses.
    - "Preference relation" is essentially the same concept as "preference."
    - "Space of preferences" is the space consisting of all possible orderings.
    - "Preference profile" is a collection of "preferences," one per dataset.

    If this is correct, I suggest rephrasing Definition A.1 for greater clarity and moving at least part of it to the main text.
* Once again, the section/subsection titles do not accurately reflect the content:
    * Section 4.1 introduces "preference profiles" but does not explore the "geometry" of preferences, despite the section's title.
    * The title of Section 4.3, "Mapping Stability to Cohomological Obstruction," is also vague and hard to follow. What exactly do you mean by "mapping" in this context?
* **Cohomology Construction in Section 4.2**: I struggled to follow the necessary background for understanding the material in Section 4.2. Unfortunately, Appendix A does not provide enough clarity either. I had to conduct additional reading to fully grasp the concepts. It would be helpful if the authors provided a more intuitive explanation, emphasizing:
    - The motivation behind the cohomology construction.
    - Why coboundaries correspond to consistent preferences.
    - Why the cohomology group captures information about Condorcet cycles.
* **Proof of Proposition 4.4**: The sentence of the proof at the end of page 4 is rather vague and unclear. Could you please make it more precise?

**Section 5**
* **Section 5.1.1**:
    * **(Confusion)** Proposition 5.1 indicates that bagging reduces cohomological obstruction in the aggregate ordinal preferences, but the aggregated ordinal preference cochain $C_P$ is only one of the cochains. Does this mean the result does not provide a general reduction in $H^1$? Clarity on this point would be helpful.

**Section 6**
* **(Confusion)** Could you elaborate on why minimizing $\mathcal{L}_{cocycle}$ reduces $H^1$? The one-line explanation provided is not sufficiently clear.

**Section 7**
* Could you provide more experimental details in the Appendix? As it stands, the setup of the experiments is not detailed enough to permit reproducibility.

**Appendix A**
* Generally, I think more background information is needed to make Appendix A self-contained and accessible to readers without prior familiarity with topological social choice theory.
    * An intuitive explanation of Theorem A.2 would be particularly helpful given its importance to this work. Specifically:
        - What is the topology on the space of preferences $\mathcal{P}^N$ that defines continuity in Theorem A.2?
        - How should we interpret "continuous aggregation rule" in this context?
    * Additionally, the ordering of the subsections in Appendix A could be improved. For example, the cohomology group (Lemma A.3) is currently introduced before the definition of cochains and the coboundary operator (Section A.4.1), which makes the presentation harder to follow.

---

> ### Author Response · Authors · 2025-10-02
>
> We thank the reviewer for recognizing the novelty of the connection and for the detailed feedback on the presentation.
>
> **W1: Writing Clarity and Structure.**
> We have significantly reorganized the paper to improve clarity and flow.
>
> * **Intuitive Overview:** We have moved the intuitive analogy ("datasets as voters," "models as candidates") to the Introduction.
> * **Section Titles and Flow (Sec. 3 and 4):** We revised the structure of Sections 3 and 4. Section 4 now clearly introduces the Combinatorial Hodge Theory framework. The definitions are presented upfront before being used.
> * **Terminology:** We have ensured terminology (Preference Profile, etc.) is clearly defined in Section 4.1.
>
> **W2: Insufficient Mathematical Background (Appendix A and Sec. 4.2).**
> We have addressed the accessibility issues.
>
> * **Section 4.2 (now 4.1–4.3):** We have replaced the abstract topological description with the concrete machinery of Combinatorial Hodge Theory (flows on graphs). This provides a rigorous yet more accessible foundation, clearly explaining why coboundaries correspond to consistent preferences (Def. 4.7) and how the Hodge decomposition captures cycles (Theorem 4.10).
> * **Appendix A:** We have reorganized Appendix A. We also added the definition of the topology on the space of preferences $\mathcal{P}$ used in Chichilnisky's Theorem (Appendix A.2.1) to ensure the paper is self-contained.
>
> **W3: Informality of Propositions/Proofs.**
> We have improved the rigor of the theoretical claims.
>
> * **Prop. 4.4 (now 4.13):** We formalized the proof by introducing the rigorous definition of "Strong Structural Inconsistency" (Def. 4.12) and updating the proof to handle general metric spaces rigorously using the triangle inequality.
> * **Prop. 5.1 (now Interpretation 5.4):** We have replaced the previous heuristic argument about bagging with a clear interpretation that rigorously distinguishes between statistical variance reduction and structural obstruction resolution. This interpretation is now strongly supported by the revised Experiments 1 and 2.
>
> **RC: Proposition 5.1 and Aggregated Ordinal Preferences.**
> The reviewer asked if the result regarding bagging (previously Prop. 5.1) only applies to the specific aggregated cochain $C_{P}^{\text{ord}}$. Yes, the analysis of structural obstructions (Condorcet cycles) mathematically requires the use of ordinal aggregation (like Pairwise Majority Vote), as cardinal aggregation always yields a cycle-free result (Section 4.2). Our framework uses $C_{P}^{\text{ord}}$ as the diagnostic tool to determine the source of instability.
>
> **RC: Section 6.** We clarified in Section 6.1 of the paper why minimizing the cocycle loss directly reduces the structural obstruction. Please see the paper (Sec. 6.1) for the precise mathematical explanation.
>
> **RC: Experimental Details (Reproducibility).**
> We have added a new Appendix B detailing the experimental setups, hyperparameters, and methodologies for all experiments to ensure reproducibility. In addition, as before, we provide all the code (updated) to reproduce our experiments, attached as a zip file to this submission.

---

> > ### Comment · Reviewer_tpE2 · 2025-10-08
> > **replies and further concerns**
> >
> > Thank you for making the changes that have enhanced the paper. While I agree that it has improved compared to the previous version, I believe there is still room for further refinement, particularly in improving the mathematical clarity and in providing a more thorough explanation of the concepts introduced. I have the following concerns and questions, and I would appreciate it if the authors could address them:
> >
> > 1. It seems to me that the paper may have downplayed the importance of prior work in social choice theory within the proposed framework, making it somewhat difficult to distinguish which aspects of the paper represent novel contributions and which are adaptations of prior work. Specifically, while the authors cite relevant prior works at the beginning of Section 4 and in Appendix A, the acknowledgments of these works feel somewhat vague. For instance, in Section 4.1, when discussing the definitions of Preference Profiles, the construction of cochains, ordinal aggregation, and cyclic constructions, it seems that many of these constructions are directly borrowed from social choice theory rather than being original contributions of this paper. However, the citations and context provided for these ideas seem insufficient. It would be helpful for the authors to offer more detailed background on when and why these concepts were originally introduced, clarify how this paper builds on prior work, and identify specific ways in which the proposed methods contribute novel insights or improvements. If there are multiple construction options in the literature, it would also be valuable to explain why these particular ones were chosen to study algorithmic stability.
> >
> > 2. The paper focuses on one particular aggregation method (ordinal aggregation), which corresponds to a specific element of the cochains introduced in Definition 4.2. This is somewhat confusing to me: what about other aggregation rules (i.e., other 1-cochains)? Is it possible that while the ordinal preference flow has a small cyclical component, there exists another cochain with a large cyclical component that could still lead to algorithmic instability?
> >
> > 3. Regarding Section 5.1.2, I am still confused about this section, even after the authors revised it. Specifically, in the term $L_{glue}$, are the h_i's and h_j's different models within the ensemble? The authors describe the ensemble members as "voters" here, and they state that this loss term enforces consensus among the ensemble members. However, in earlier parts of the paper, weren’t the "voters" defined as different datasets? In Section 4.1, it is noted that the voters are "individual data points in an ensemble context". What does this mean? This distinction remains very unclear to me.

---

> > > ### Author Response · Authors · 2025-10-09
> > > **further clarifications**
> > >
> > > We have further revised the manuscript to address the reviewer’s concerns.
> > > - **Acknowledgment of Prior Work:** Clearer attributions to foundational concepts from social choice theory and combinatorial Hodge theory are now included in Sections 4.1 and A.1. We have kept this discussion concise in the interest of the paper's overall length but would be happy to expand it if the reviewer feels it is warranted.
> > > - **Our Novel Contributions:** The paper’s novelty lies in applying these mathematical foundations to the problem of *algorithmic stability*. Specifically:
> > >   1. Identifying the cyclical component $(\(\|C_{\text{cycle}}\|\))$ as the mathematical obstruction underlying structural instability.
> > >   2. Using $\(\|C_{\text{cycle}}\|\)$ to distinguish structural from statistical instability (Experiments 1, 2).
> > >   3. Unifying existing stability methods (bagging, inflation, regularization) under a single cohomological framework (Section 5).
> > >   4. Empirically showing $\(\|C_{\text{cycle}}\|\approx0\)$ in standard tasks but $\(\|C_{\text{cycle}}\|>0\)$ in fairness-sensitive settings (Experiment 5).
> > >   5. Introducing and validating obstruction-aware regularization (Sections 6 and 7.3).
> > >
> > > We updated **Section 1** to list these contributions explicitly, strengthened **Section 2.3** to better separate adaptation from novelty, and clarified the mathematical adaptation at the start of **Section 4**.
> > >
> > > The reviewer asked why we focus solely on the ordinal preference flow (a specific 1-cochain) and whether other aggregation rules could reveal different instabilities.  Our focus on ordinal aggregation via Pairwise Majority Vote (PMV) is *mathematically necessary* for detecting the structural inconsistencies (Condorcet cycles) relevant to stability.
> > >
> > > - **Why not Cardinal Aggregation?**
> > >   The main alternative—cardinal aggregation (averaging loss differences)—*cannot* detect Condorcet cycles. As clarified in Section 4.2, cardinal aggregation always yields a gradient flow $\(C_{\text{cycle}}=0\$) because it is the gradient of the average loss. Formally, Chichilnisky (1980) shows that aggregation paradoxes arise only in the *ordinal* case, where the preference space is non-contractible. In contrast, *cardinal* aggregation operates in a linear, contractible space and thus faces no topological obstruction. In our framework, the ordinal rule defines a non-exact 1-cochain with a residual cyclical component, whereas the cardinal rule is exact (cycle-free). Hence, only ordinal aggregation can exhibit instability through such cycles.
> > >
> > > - **Why PMV is Canonical:**
> > >   The Condorcet paradox is inherently ordinal (“a majority prefers A > B, B > C, C > A”). To connect algorithmic stability with this classical structure, we must use an aggregation rule that respects ordinal preference relations. PMV is the canonical choice in both social choice theory and Hodge-theoretic rank aggregation.
> > >
> > > - **Other 1-cochains:**
> > >   While arbitrary 1-cochains can be defined, they need not represent meaningful preference aggregation. The instability we analyze arises from the data’s intrinsic preference structure, which PMV captures faithfully.
> > >
> > > We expanded **Section 4.2** and added **Remark 4.8**, emphasizing that PMV is the necessary and canonical choice for analyzing structural inconsistencies due to preference cycles.
> > >
> > > We resolved inconsistencies in the definition of “voters” between theoretical and applied sections:
> > >
> > > - **Section 4.1 (Theory):** “Voters” denote entities expressing preferences over hypotheses (candidates). In practice, this means the *data distribution* $\(\mathcal{D}\)$, approximated by individual data points serving as empirical voters (as clarified in Experiment 2).
> > > - **Section 5.1.2 (Regularization):** Ensemble members were mistakenly described as “voters.” This has been corrected: regularization aligns models’ induced preferences over data, thereby reducing the cyclical component.
> > > - **Section 6.1 (L\_{cocycle}):** We clarified that this section shifts perspective—from analyzing preference cycles (datasets voting on models) to enforcing geometric consistency among model representations.

---

### Review · Reviewer_gu4X · 2025-09-29

**Summary Of Contributions:**

__Summary:__
This paper presents the robustness of algorithms from the viewpoint of algebraic topology. In particular, the authors characterize the lack of algorithmic stability in terms of the existence of a non-trivial first cohomology group of the 1D cochain complex derived from the utility function. The authors demonstrate that standard techniques to increase robustness, such as bagging and regularization, can be understood from this perspective -- as reducing this topological obstruction term.

__Strengths:__
- The perspective presented in this paper is unique and interesting.

__Weaknesses:__
- I find some of the arguments in the paper to be unclear and could be supported with more details (see below in __Requested Changes__).
- The proposed obstruction-aware ensembling loss in Definition 6.1 has cubic complexity (due to the triple sum). This doesn't seem to be very practical in real-world settings.
- I find the experiments to be quite weak overall and unconvincing. All of them are on simple synthetic datasets that are quite far from realistic ML tasks. Explanations are ambiguous and there's not enough details to undetstand what it is doing. More concretely,
    - Experiment 1 is confusing to me; what is the purpose if the proposed solution to the instability is a method that return _all_ hypotheses?
    - In Experiment 2, how reliable is the result if the magnitudes observed of the obstruction term are of the order of magnitude of machine precision? Couldn't the observed "improvement" be just noise?
    - I don't understand what Experiment 3 is showing. What is the hypothesis space and the model that is being trained? When discussing "improvements", it is improvements with respect to what?
- Methods such as bagging and regularization are typically used to increase generalizability (practically, this is robustness on a separate held-out validation set). I don't see the immediate link between this and the training data stability that is considered in this work.

**Audience:**

Yes

**Audience Explanation:**

If presented more convincingly, I believe there will be some interest in the direction of the work; of trying to understand model instability from a topological viewpoint.

**Broader Impact Concerns:**

The work is mostly theoretical. To the best of my knowledge, there are no negative ethical implications.

**Claims And Evidence:**

No

**Claims Explanation:**

I find the presentation in the paper to be vague overall and do not provide enough convincing evidence about its claims. Theoretically, some of the steps in the proof are brushed over, relying more often than not on plain language arguments than concrete mathematical claims. Additionally, I find the experiments to be too simplistic, detatched from real-world ML problems, and lacking in concrete details.

**Requested Changes:**

Overall, I find the paper to be very mixed in terms of mathematical rigour. Some concepts are introduced precisely, yet others are vague (or perhaps, at least to those who are not familiar with social choice theory). I would encourage the authors to be more precise throughout. For example:
- Could you please state the construction of the 0-cochains, 1-cochains, and the coboundary operator more concretely in Section 4.2? For example, what is "pairwise comparison" and what does it mean to "transform utility into induced comparisons".
- How is $\succ$ defined without reference to a dataset? In Definition 3.2 $\succ\_S$ is defined as a preference ordering with respect to a dataset $S$, but in Section 4, a "global" consensus ranking $\succ\_G$ is introduced without a precise definition, and thereafter, $\succ$ without reference to a specific dataset is used, with no mention of what it means.

Some of the arguments in the proofs are not clear, and the mathematical rigor is questionable. For example:
- In the proof of Proposition 4.4, can you be more mathematically precise about the statement "If the stability constraint $\epsilon$ is too tight relative to the separation of the hypothesis and the geometry of $\mathcal{D}$, the algorithm cannot transition between the preferred hypotheses $h\_i$, while remaining stable".
- In the proof of Proposition 5.1, I don't understand why "Instability increases the probability that the local preferences form a cycle".
- The statement about the bagged learner decreasing the variance of $V_{ij}$ in Proposition 5.1 is not very rigorous.
- As before, I don't understand why "the occurence of localized inconsistencies (local cycles) decreases" as "the preferences become more stable".
- The statement of Proposition 5.1 talks only about decreasing the topological obstruction term, but algorithmic interpretation is provided only in the cases when the term is either trivial (a global preference exists) or non-trivial (a structural inconsistency exists). What does are the algorithmic implications in the non-trivial case, when the obstruction term is large or small?

---

> ### Author Response · Authors · 2025-10-02
>
> We thank the reviewer for their detailed assessment and very constructive criticisms.
>
> **W1/E3: Weak Experiments and Near-Zero Magnitudes in Exp 2.** The reviewer found the experiments unconvincing and questioned the reliability of the near-zero cyclical norms ($10^{-16}$) in Exp 2. This critique led to a crucial refinement of our framework.
>
> *   *Interpretation of Near-Zero Norms (Exp 2):* We confirm that these norms are indeed indistinguishable from machine precision. We have revised Section 7.2 to reflect this honestly. This finding demonstrates that instability in these standard scenarios is statistical (due to variance), not structural ($C_{cycle} \approx 0$). We added empirical stability metrics in Table 2, showing high instability coexisting with $C_{cycle} \approx 0$. As predicted by our framework when instability is statistical, bagging successfully reduces it (e.g., from 0.4610 to 0.1963 in Scenario B).
> *   *Bagging vs. Structural Instability (Exp 1):* We revised Experiment 1 (Section 7.1) to test bagging against a pure structural obstruction (Condorcet cycle). Bagging fails completely (Stability 0.3432), while inflation succeeds (Stability 1.0000).
> *   **(NEW) Real-World Structural Obstructions (Exp 5):** To address whether significant structural obstructions occur in practice, we added a new Experiment 5 (Section 7.5) on fairness-constrained model selection using the UCI Adult dataset. The results are striking: we found $||C_{cycle}|| \approx 0.857$, which is 15 orders of magnitude larger than in standard tasks. This confirms that structural obstructions are a real phenomenon in domains with conflicting objectives, validating the relevance of our framework and providing a novel topological quantification of fairness impossibility theorems.
>
> **W1: Unclear Arguments and Rigor (Prop 4.4 and 5.1).** We have significantly improved the rigor.
> *   *Prop 4.4 (Now 4.13):* We formalized the vague language. The proposition now relies on the rigorous definition of "Strong Structural Inconsistency" (Def 4.12), and the proof has been updated to rigorously address the general metric space case using the triangle inequality.
> *   *Prop 5.1 (Now Interpretation 5.4):* We have replaced the previous heuristic argument about bagging with a clear interpretation supported by the evidence in Exp 1, 2, and 5, clarifying when and why bagging works.
>
> **W2: Cubic Complexity $O(N^3)$ of Proposed Loss.** This is a valid concern. We have addressed this by proposing a stochastic approximation $O(K)$ (Section 6.1). We added a new Experiment 4 (Section 7.4) demonstrating that the stochastic version is scalable (3.34x speedup) and effective (reduces defect by 5 orders of magnitude).
>
> **W3: Experiments on Simple Synthetic Datasets.** We acknowledge this limitation. We have added a justification at the start of Section 7 explaining that controlled settings (synthetic or conflict-engineered real data) are scientifically necessary to isolate and study the effects of Condorcet cycles. Crucially, we have now included Experiment 5 (Section 7.5), which validates the framework on the real-world UCI Adult dataset, demonstrating the emergence of structural obstructions in fairness scenarios and moving beyond purely synthetic validation.
>
> **W3: Exp 1 Confusion (Returning all hypotheses).** The goal of inflated operators is to achieve stability by capturing ambiguity when a single stable choice does not exist. In a Condorcet cycle, the ambiguity involves all candidates. Exp 1 confirms that returning the full set is the only way to achieve perfect stability (1.0000) in this scenario. We clarified this in Section 7.1.
>
> **W4: Link between Training Stability and Generalizability.** The foundational work of Bousquet & Elisseeff (2002) established that algorithmic stability is a key condition for deriving generalization bounds. We have clarified this connection in the first sentence of the Introduction.
>
> **RC1: Precise Definitions (Cochains, Coboundary Operator).** We have completely revised Section 4, pivoting to Combinatorial Hodge Theory. Sections 4.1 and 4.2 now provide concrete definitions for cochains, flows, the coboundary operator, and the crucial distinction between Cardinal and Ordinal aggregation.
>
> We have revised the manuscript to emphasize that our core contribution is theoretical unification, rather than the provision of immediate practical tools.

---

### Author Response · Authors · 2025-10-02
**summary of main changes**

We sincerely thank the reviewers for their insightful and constructive feedback. The reviews raised substantial concerns regarding mathematical rigor, experimental validation, and clarity. We have undertaken a major revision to address these concerns, significantly strengthening the paper.

Key Revisions:

*   **Enhanced Mathematical Rigor via Combinatorial Hodge Theory:** Addressing concerns from all reviewers regarding rigor and clarity (EcaC, tpE2, gu4X), we have pivoted the presentation to the concrete machinery of Combinatorial Hodge Theory (Section 4). This grounds the framework in rigorous analysis of flows on graphs, minimizing abstract jargon. We formalized Proposition 4.13 (Instability Induced by Obstructions) by introducing the "Strong Structural Inconsistency" definition (Def 4.12) and generalizing the proof to handle general metric spaces, resolving the statement-proof mismatch (EcaC).

*   **Refined Framework and Clarified Contributions: Unifying Stability via Structural vs. Statistical Distinction:** The most significant conceptual refinement, driven by the reviewers' experimental critiques (gu4X) and questions regarding practical relevance, is the rigorous distinction between Structural Instability ($C_{cycle} > 0$) and Statistical Instability ($C_{cycle} \approx 0$). We have reframed the paper (Sections 1, 5, 8) to emphasize the theoretical unification this provides. The framework explains *why* and *when* different stabilization methods are effective based on the underlying mathematical structure of the instability.

*   **Strengthened Experimental Validation (From Synthetic Cases to Real-World Phenomena):** We have substantially revised our experiments to validate this refined framework.
    *   *Interpreting Near-Zero Norms:* Reviewer gu4X questioned the reliability of the near-zero norms in Exp 2. We confirm these norms are indistinguishable from machine precision. This crucial finding demonstrates that instability in these standard scenarios is *statistical*, not structural (Section 7.2), explaining the empirical success of bagging in these domains.
    *   *Validating the Distinction:* We updated Experiment 1 (Section 7.1) to show that bagging *fails* (Stability 0.34) against a pure structural obstruction (Condorcet cycle), while inflation succeeds (Stability 1.00). Conversely, Experiment 2 shows that when instability is statistical ($C_{cycle} \approx 0$), bagging *succeeds*. These results validate our unified view (Interpretation 5.4).
    *   **(NEW) Demonstration of Real-World Structural Obstructions (Fairness):** To address whether structural obstructions occur in practice, we added a new Experiment 5 (Section 7.5) on fairness-constrained model selection using the UCI Adult dataset. We found significant structural obstructions ($||C_{cycle}|| \approx 0.86$), 15 orders of magnitude larger than in standard tasks. This confirms the relevance of our framework in domains with conflicting objectives and provides a novel topological characterization and quantification of fairness impossibility theorems.

*   **Addressed Scalability and Practicality:** We addressed the $O(N^3)$ complexity concern (gu4X) regarding our proposed $\mathcal{L}_{cocycle}$. We introduced a stochastic approximation $O(K)$ (Section 6.1) and added a new Experiment 4 (Section 7.4) demonstrating its scalability (3.34x speedup) and effectiveness.

We believe these revisions have resulted in a significantly more rigorous paper.

---

### Author Response · Authors · 2025-11-16
**Revision to improve clarity**

After reflecting on what may have prevented reviewer EcaC, who provided the most recent comments and questions, from fully understanding the paper, and after attempting to address his concerns directly, we concluded that the paper indeed lacked clarity. In particular, it was not obvious—except perhaps to the authors—that the stability notion used in the paper is more general than usually assumed. Additionally, while numerous, the experiments were not well designed to complement the theoretical framework. We have therefore undertaken a major revision to improve mathematical clarity and reduce the number of computational experiments while ensuring they remain strongly motivated. We hope this presentation is clearer and thank the reviewers for pointing out the paper's deficiencies and for questions that helped improve its clarity.

---

### Decision · Action_Editor_A5vx · 2025-12-11

**Recommendation:** Accept as is

**Audience:**

Yes

**Audience Explanation:**

The topic is somewhat niche -- geometric insights into algorithmic stability using algebraic topology and social choice theory -- but reviewers found it interesting, and this could be interesting to others in the learning theory community.

**Claims And Evidence:**

Yes

**Claims Explanation:**

The initial manuscript was lacking some clarity in the mathematical definitions needed to understand the main results, but after discussions with reviewers, this has been improved.